# Overcoming strength-ductility tradeoff with high pressure thermal treatment

Yao Tang [1,2], Haikuo Wang[1] ✉, Xiaoping Ouyang[1,3] ✉, Chao Wang[1], Qishan Huang [2], Qingkun Zhao [2], Xiaochun Liu [4], Qi Zhu [5], Zhiqiang Hou[1], Jiakun Wu[1], Zhicai Zhang[1], Hao Li[1], Yikan Yang[1], Wei Yang[2], Huajian Gao [5,6,7] ✉ & Haofei Zhou [2] ✉

Conventional material processing approaches often achieve strengthening of materials at the cost of reduced ductility. Here, we show that high-pressure and high-temperature (HPHT) treatment can help overcome the strength-ductility trade-off in structural materials. We report an initially strong-yet-brittle eutectic high entropy alloy simultaneously doubling its strength to 1150 MPa and its tensile ductility to 36% after the HPHT treatment. Such strength-ductility synergy is attributed to the HPHT-induced formation of a hierarchically patterned microstructure with coherent interfaces, which promotes multiple deformation mechanisms, including dislocations, stacking faults, microbands and deformation twins, at multiple length scales. More importantly, the HPHT-induced microstructure helps relieve stress concentration at the interfaces, thereby arresting interfacial cracking commonly observed in traditional eutectic high entropy alloys. These findings suggest a new direction of research in employing HPHT techniques to help develop next generation structural materials.

Various industrial and advanced functional applications require materials with high strength and ductility. Unfortunately, conventional processing strategies usually attain high material strength at the expense of deteriorated ductility[1–3], whilst developing structures with substantial improvement of both strength and ductility is highly challenging. Recent successes in synthesizing superhard materials[4–6] have made the high-pressure and high-temperature (HPHT) treatment[7,8] promising to achieve extreme properties in materials. The combination of pressure and temperature offers the great prospect for altering the microstructure of structural materials that would permit overcoming the strength-ductility trade-off. Given the vast space of control parameters involved in the HPHT treatment and the often

complex microstructural evolution under extreme pressures and temperatures[9,10], it remains difficult to predict and interpret the mechanical behaviors of HPHT-treated structural materials, especially the abnormal kinetics and thermodynamics of non-equilibrium interfaces induced by the HPHT treatment[11,12].

Eutectic high entropy alloys (EHEA) represent a promising class of multi-principal-element alloys featured by a hierarchical microstructure of dual-phase lamellae colonies, which offers great potential for achieving superior mechanical properties that are unmatched by conventional metallic materials[13]. However, the room-temperature brittleness and limited tensile ductility of these alloys hinder their usage in practical applications, despite a wide range of microstructural

[1]Center for High Pressure Science and Technology, College of Energy Engineering, Zhejiang University, Hangzhou, China. [2]State Key Laboratory of Fluid Power and Mechatronic Systems, Center for X-Mechanics, Department of Engineering Mechanics, Zhejiang University, Hangzhou, China. [3]School of Materials Science and Engineering, Xiangtan University, Xiangtan, China. [4]Institute of Metals, College of Material Science and Engineering, Changsha University of Science and Technology, Changsha, China. [5]School of Mechanical and Aerospace Engineering, College of Engineering, Nanyang Technological University, Singapore 639798, Singapore. [6]Institute of High Performance Computing, A*STAR, Singapore 138632, Singapore. [7]Mechano-X Institute, Applied Mechanics Laboratory, Department of Engineering Mechanics, Tsinghua University, 100084 Beijing, China. ✉ e-mail: haikuo.wang@zju.edu.cn; oyxp2003@aliyun.com; gao.huajian@tsinghua.edu.cn; haofei_zhou@zju.edu.cn

design strategies and associated processing techniques have been proposed[14,15]. Due to the prevalent interfacial structures in eutectic high entropy alloys, pronounced dislocation pile-ups are routinely developed in the vicinity of internal boundaries, where local stress concentration can emerge, leading to the onset of micro-cracking and catastrophic damage during mechanical straining[16,17].

Here, we demonstrate that, with HPHT treatment in a self-designed hexahedron-anvil-press apparatus, the dual-phase lamellae structure frequently observed in eutectic high entropy alloys[14,18] can be transformed to a hierarchically patterned microstructure with coherent interfaces. Tensile tests at ambient temperature demonstrate that the HPHT-treated specimens exhibit a high uniform tensile elongation (~36%) and greatly improved fracture strength (~1150 MPa), overcoming the typical brittle interfacial fracture mode in conventional eutectic high entropy alloys. Multiple deformation mechanisms, including dislocations, stacking faults, microbands and deformation twins, are observed at multiple length scales, as the applied strain increases. Isolated microcracks are uniformly distributed near the fracture surface in the HPHT-treated samples, in contrast to the catastrophic fracture in those without HPHT treatment. We further demonstrate that the HPHT treatment can be easily extended to other alloy systems for synergistic improvement of strength and ductility.

## Results

We used the homogenized $Al_{0.7}CoCrFeNi$ (molar ratio) alloy[18] (hereafter termed Ho alloy) with a composition of $Al_{14.8}Co_{21.3}Cr_{21.3}Fe_{21.3}Ni_{21.3}$ (at. %) and the typical dual-phase lamellae microstructure (Fig. 1a, b) as the precursor material for HPHT treatment. The corresponding EBSD phase map (Fig. 1c) clearly reveals the dual-phase lamellae consisting of alternating face-centered cubic (FCC) layers and body-centered cubic (BCC) layers. After HPHT treatment (Fig. 1f) at a pressure of 6 GPa and a temperature of 1473 K (more details see Methods), we confirmed that there is no change in the phase constitutions with X-ray diffraction (XRD) (Fig. 1g), where the corresponding peak positions of the HPHT-treated alloy are shifted slightly rightward due to densification. Notably, the HPHT-treated alloy exhibits a hierarchically patterned microstructure (Fig. 1h) consisting of hexagonal-like structural units (Fig. 1i). The corresponding EBSD phase map (Fig. 1j) reveals that the interior of each structural unit is composed of a mixture of FCC and BCC phases, while different units are interconnected by the FCC phase. The volume fractions of FCC and BCC phases in the HPHT-treated alloy were determined by EBSD to be ~69% and ~31%, nearly identical to those in the Ho alloy (~68% and ~32%). The elemental distribution maps of the HPHT-treated alloy (Fig. 1m–q) indicate that the FCC phase is relatively depleted in Al and Ni elements, with a composition of $Al_9Co_{22}Cr_{28}Fe_{24}Ni_{17}$ (at. %), while the BCC phase is rich in Al and Ni, with a composition of $Al_{30}Co_{19}Cr_9Fe_{13}Ni_{29}$ (at. %). Despite of the substantial change in microstructural pattern, the compositions of the FCC and BCC phases in the alloy before (Supplementary Fig. 1) and after HPHT treatment show almost no difference. Transmission electron microscopy (TEM) was used to better characterize the microstructures of the HPHT-treated alloy (Fig. 1k) and Ho alloy (Fig. 1d). The presence of super-lattice spots in the selected area electron diffraction (SAED) patterns suggests an ordered B2 structure for the BCC phase. No other phases were detected in both alloys. A high-resolution TEM image of the interface region between the two phases (highlighted by the white dashed circle in Fig. 1k) in the HPHT-treated alloy reveals a coherent interface structure (Fig. 1l and Supplementary Fig. 3). The inset in Fig. 1l presents the corresponding fast Fourier transform (FFT) patterns, in which the diffraction spots with yellow and blue circles are the [100] crystal zone axis of the FCC and B2 phases, respectively. The orientation relationship between the two phases is confirmed to be (01-1)B2//(−111)FCC, which meets the classical Kurdjumov−Sachs (KS) relationship[19], suggesting a high coherency between the FCC phase

and B2 phase in the HPHT-treated alloy. Further analysis shows that the two phases possess a mismatch factor of merely 0.004. Such coherent FCC/B2 interface in the HPHT-treated alloy is clearly distinguished from the corresponding interface in the Ho alloy, where a transition layer with a thickness of ~10 nm between the FCC and B2 phases was identified in the lamellae structure, as shown in Fig. 1e and Supplementary Fig. 2. Similar incoherent interface has been frequently reported in eutectic high entropy alloys and recognized as the origin of strain localization[17,20]. The hierarchically patterned structure with coherent interfaces modulated through HPHT treatment is expected to have a major impact on the mechanical response of the HPHT-treated alloy, which is generally inaccessible to conventional materials processing techniques.

Figure 2 displays the tensile properties of the HPHT-treated alloy and a number of comparison alloys at room temperature. The HPHT-treated alloy exhibits nearly 100% enhancement in both strength (~1150 MPa) and tensile ductility (36%) compared with the Ho alloy. Such HPHT-induced synergistic enhancement in strength and ductility is not observed in our experiments with conventional hot isostatic pressing treatment (Supplementary Fig. 5) and thermal-mechanical treatment (Supplementary Fig. 6). The Ashby map of ultimate tensile strength versus elongation demonstrates the superior strength-ductility synergy of the HPHT-treated alloy, compared with other eutectic high entropy alloys reported in the literature[18,21–38] (Fig. 2b). For reference, the tensile stress-strain curves of the as-cast alloy specimens with (marked as Ac+HPHT alloy) and without (Ac alloy) HPHT treatment are also presented in Fig. 2. The former exhibits a remarkable improvement in ductility compared to the Ac alloy which fails by brittle cleavage, again demonstrating the benefit of HPHT treatment. The reduction in strength of the Ac samples can be attributed to the change in the microstructure with HPHT treatment (Supplementary Note 3). The corresponding true stress-strain curves of the Ac, Ac+HPHT, Ho and Ho+HPHT alloys also exhibit remarkable differences in fracture stress and strain (the inset shown in Fig. 2a). The Ac+HPHT alloy exhibits a remarkable improvement in true fracture strain compared to the Ac alloy, while the true fracture strength is almost the same. The Ho alloy only exhibits a true fracture strain of ~14.7%, while the HPHT alloy doubles the fracture strain to ~30.7%. More importantly, the HPHT alloy exhibits a high true fracture stress of ~1570 MPa, which is considerably higher than those of Ac, Ac+HPHT and Ho alloys, respectively. The corresponding work hardening rate curves of these alloys have been calculated (Supplementary Fig. 7). In comparison with the Ac, Ac+HPHT and Ho alloys, the work-hardening capability of HPHT-treated alloy is much more pronounced and stable up to the tensile plastic instability region. The differences among strain-hardening rate curves strongly indicate the different deformation behaviors of these alloys during plastic deformation. The properties of HPHT-treated alloys can be further tuned by changing the applied pressures and temperatures (Supplementary Fig. 9), depending on the structural stability of the hierarchically patterned microstructures.

The crack surface morphology of the fractured samples was examined to explain the exceptional tensile properties of the HPHT-treated alloy. Different from the elongated cracks with large width along the interfaces in the fractured Ho alloy, a high density of microcracks and a more uniform crack surface morphology can be observed in the HPHT-treated alloy (Supplementary Fig. 10). Specifically, numerous isolated microcracks cut through the B2 phase and the fracture paths were twisted among the two phases in the HPHT-treated alloy, preventing crack coalescing into elongated voids. Besides, a number of unbroken ligaments were observed near the fracture surface, which is believed to shield crack propagation. The enhanced tensile elongation of the HPHT-treated alloy can thus be related to the delocalized damage accumulation and arrest of crack propagation.

The microstructure evolution of the tensile-deformed HPHT-treated alloy at various strain amplitudes was analyzed by EBSD

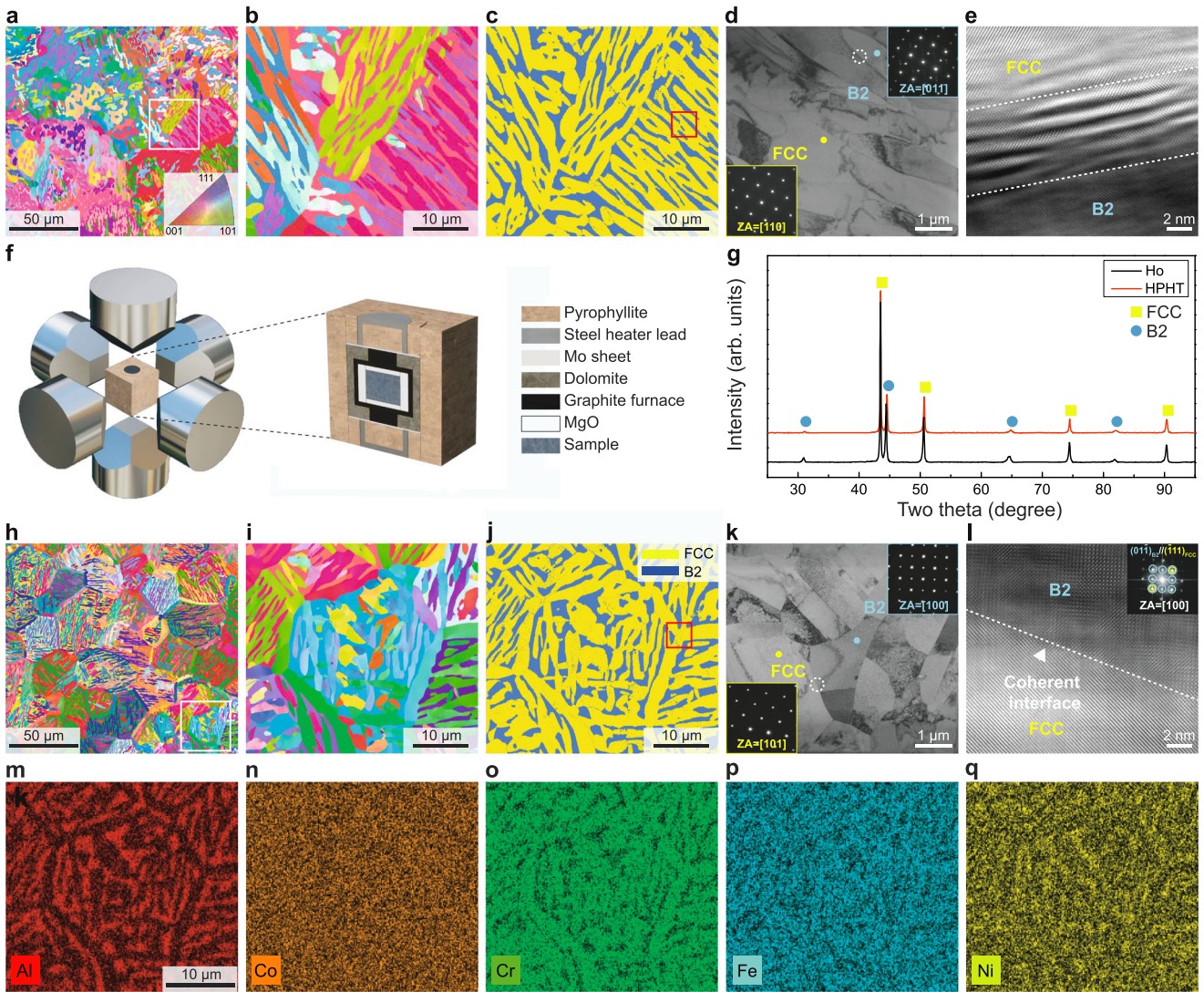

**Fig. 1 | Microstructure of Al$_{0.7}$CoCrFeNi eutectic high entropy alloys before and after HPHT treatment. a** An electron backscatter diffraction (EBSD) inverse-pole figure (IPF) map of the homogenized Al$_{0.7}$CoCrFeNi alloy (Ho alloy) before HPHT treatment, which serves here as the precursor material for HPHT treatment. **b** A magnified image of the lamellae structure (the region highlighted by the white box in **a**). **c** The EBSD phase map corresponding to b reveals the dual-phase lamellae consisting of alternating FCC phase and BCC phase in Ho alloy. **d** A bright-field transmission electron microscopy (TEM) image of the lamellae structure (the region highlighted by the red box in **c**). The insets show the related selected area electron diffraction (SAED) patterns of FCC and B2 phases (indicated by a yellow dot and a blue dot, respectively). **e** A high-resolution TEM image of the interface between the FCC and BCC phases highlighted by the white dashed circle in **d**, revealing a transition layer between the FCC phase and the B2 phase. **f** Schematic

diagram of the HPHT treatment set-up and the sample assembly part. **g** XRD patterns of Ho and HPHT-treated alloys. **h** The EBSD-IPF map of the HPHT-treated alloy with a hierarchically patterned structure. **i** Enlarged view of the composed hexagonal-like structural unit shown in **h** (the region highlighted by the white box). **j** The EBSD phase map corresponding to **i**. **k** A TEM image of the hierarchically patterned structure (the triple junction region highlighted by the red box in **j**). The insets show the corresponding selected area electron diffraction (SAED) patterns of FCC and B2 phases (indicated by a yellow dot and a blue dot, respectively). **l** A high-resolution TEM image of the interface highlighted by the white dashed circle in k confirming the interfacial coherency. The inset is the corresponding fast Fourier transform (FFT) patterns, showing a Kurdjumov–Sachs (K-S) relation between the FCC and B2 phases. **m–q** Energy-dispersive spectroscopy maps of the EBSD-mapped region shown in **i**.

(Fig. 3a). The regions at different strain amplitudes for EBSD analysis of the tensile-deformed HPHT-treated alloy are indicated in Supplementary Fig. 12. Notably, the featured hierarchically patterned microstructure is well preserved during loading, which can even be observed faintly on the fracture surface after the extended uniform elongation, indicating the high structural stability of the HPHT-treated alloy. The corresponding EBSD phase maps confirm that the dual-phase structure is maintained during tension (Fig. 3b). It is easy to identify the seriously damaged B2 phases and FCC phases at 35% strain from the EBSD phase map. The distribution of plastic deformation in the HPHT-treated alloy at different strain amplitudes can be analyzed using the kernel average misorientation (KAM) maps[39] (Fig. 3c), which can be

retrieved directly from EBSD data. At small plastic strains, the FCC phase exhibits relatively higher KAM values than the B2 phase. With increasing strain, the B2 phase possesses increasing KAM values, which helps relieve the strain localization in the deformed FCC phase. In particular, a relatively homogeneous distribution of KAM is observed at the final fracture with a plastic strain of 35%, contributing to the extra tensile elongation of the HPHT alloy. In contrast, in the Ho alloy, the B2 phase possesses much smaller KAM values than the FCC phase in the later stage of tensile deformation (Supplementary Fig. 15). The increased misorientation distribution of FCC and B2 phases with increasing strain also indicates strain delocalization in the tensile-deformed HPHT-treated alloy (inset in Fig. 3d). Since the KAM value is

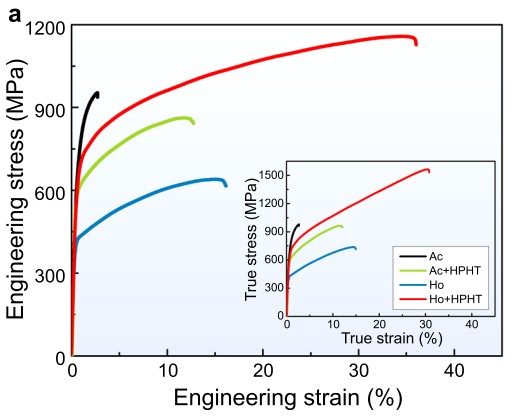

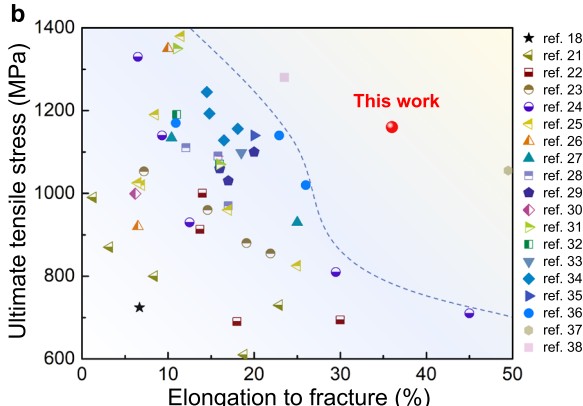

**Fig. 2 | Tensile properties of HPHT-treated alloys at room temperature.**
**a** Engineering stress-strain curves of the homogenized alloy (Ho alloy) after HPHT treatment, along with those of the Ho alloy before HPHT treatment and the as-cast alloy (Ac alloy) with and without HPHT treatment, showing that the HPHT treatment resulted in significant increases in both strength and ductility. The corresponding true stress-strain curves are shown in the inset. **b** Ashby map of the ultimate tensile stress versus uniform elongation of the HPHT-treated alloy compared to other eutectic high entropy alloys reported in the literature.

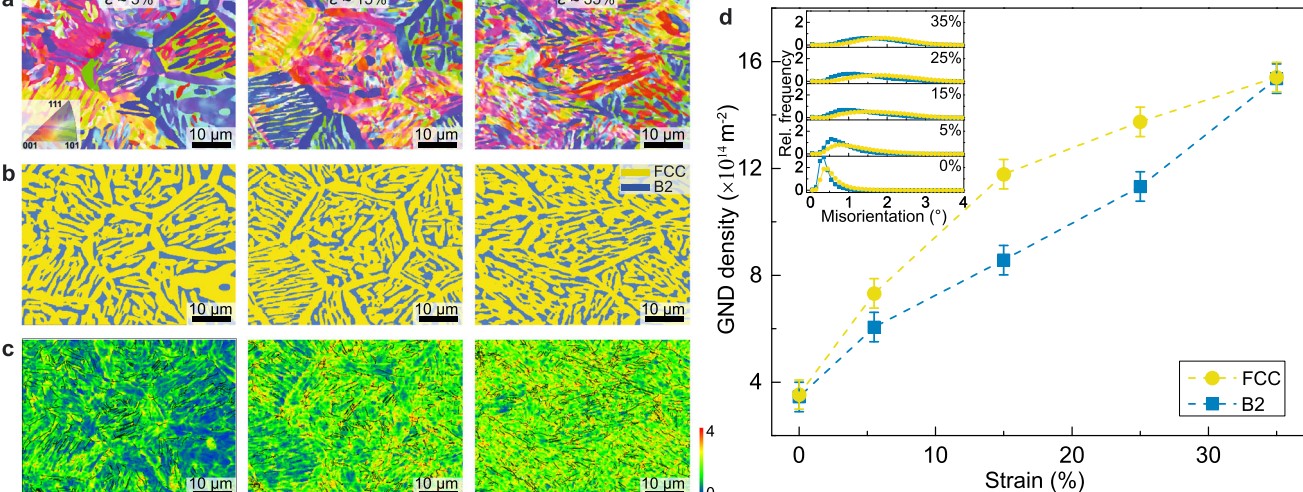

**Fig. 3 | Microstructure evolution of the HPHT-treated alloy at different strain amplitudes. a** Electron backscatter diffraction (EBSD) inverse-pole figure (IPF) maps at the strain amplitudes of 5%, 15%, and 35%. **b** The corresponding phase distribution maps. **c** The corresponding kernel average misorientation (KAM) distribution maps. **d** Variations of average geometrically necessary dislocations (GND) densities in the FCC and B2 phases, respectively. The error bars represent the corresponding standard deviation, which are obtained from 3 independent EBSD mappings on the regions with identical local strain levels. The inset shows the variation in the average misorientation of the HPHT-treated alloy with increasing plastic strain.

associated with the density of geometrically necessary dislocations (GNDs)[40], higher KAM values suggest a higher density of GNDs and more plastic strains in these regions. Therefore, GND density can be estimated from the KAM value and help us understand the difference in deformation behaviors of the HPHT-treated alloy and Ho alloy. To quantify the plastic deformation accommodation in FCC and BCC phases, we have calculated the GND maps (Supplementary Fig. 12) and average GNDs values under various strains in the HPHT-treated alloy (Fig. 3d). From 0% to 15% strains, the GND density of the FCC phase increases rapidly from $3.53 \times 10^{14}$ m$^{-2}$ to $11.78 \times 10^{14}$ m$^{-2}$, whereas the GND density of the B2 phase increases from $3.44 \times 10^{14}$ m$^{-2}$ to $8.56 \times 10^{14}$ m$^{-2}$. Upon further loading, the GND density of the B2 phase increases much faster than that of the FCC phase, eventually resulting in an identical value of GND density in both phases at 35% strain, consistent with a homogeneous plastic deformation distribution (Fig. 3c). In the Ho alloy, however, the average GND density of the B2 phase shows a severe mismatch with that of the FCC phase, indicating a highly incompatible plastic strain distribution (Supplementary Figs. 15

and 16). Moreover, the GND densities in the HPHT alloy are much higher than those in the Ho alloy, indicating improved capability of dislocation storage in the HPHT-treated alloy. In addition, the total dislocation densities measured at various strains (Supplementary Note 5) confirm the uniform deformation of the HPHT-treated alloy.

## Discussion
To further reveal the underlying deformation mechanisms, we performed electron channeling contrast imaging (ECCI) analysis at various strains for the HPHT-treated alloy (Fig. 4a–c). We have carried out five individual tests of ECCI analysis for different locations at the same strain amplitude to confirm the observed deformation characteristics of the HPHT-treated alloy. A typical triple junction region of the hierarchically patterned structure (indicated in Supplementary Fig. 13) was selected for demonstration, with magnified ECCI images presented in Fig. 4a–c. At a strain of 5%, profuse dislocation activities are observed in both FCC and B2 phases, with the dislocations extending across the FCC/B2 interfaces, indicating an early stage of synergistic deformation.

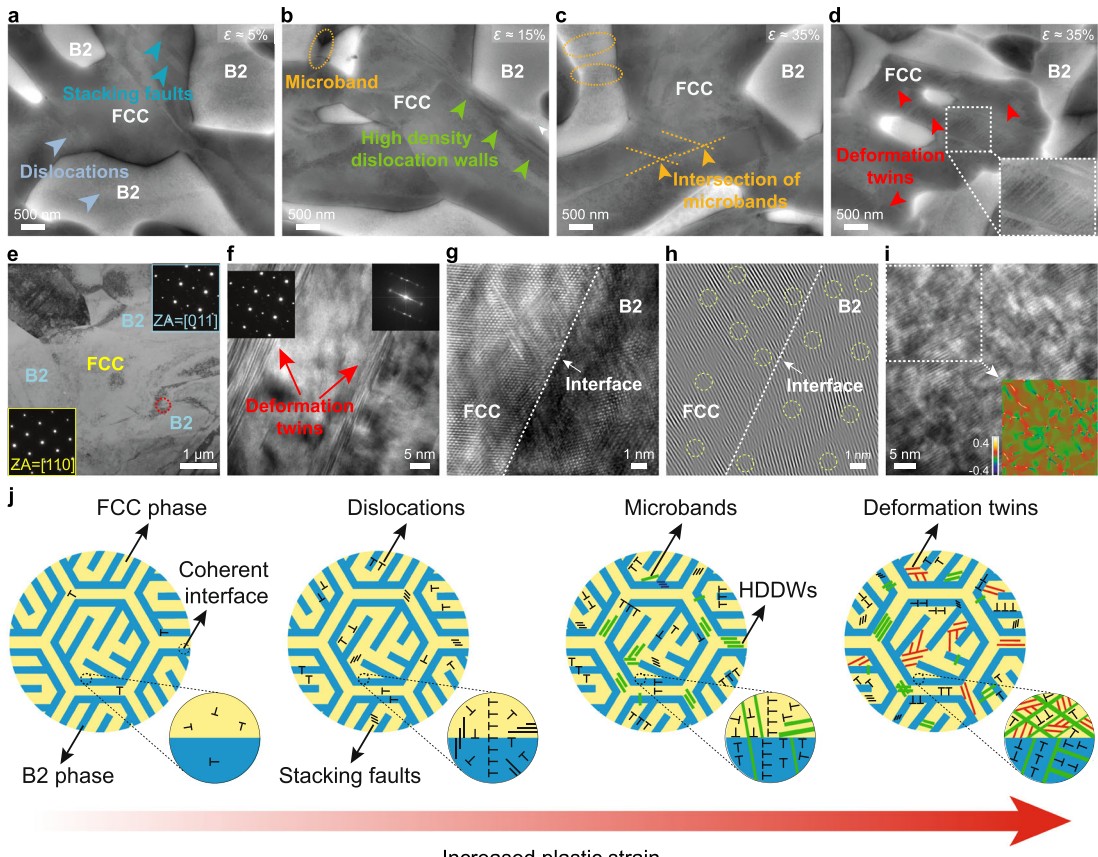

**Fig. 4 | Activation of multiple deformation mechanisms in the tensile-deformed HPHT-treated alloy. a–c** Electron channeling contrast imaging (ECCI) analysis revealing the evolution of the deformation substructure in the HPHT-treated alloy at the strain amplitudes of 5%, 15%, and 35%. **d** Multiple deformation twins activated in the HPHT-treated alloy deformed to 35% plastic strain. The inset shows a magnified image of the deformation twins. **e** Bright-field transmission electron microscopy (TEM) image of the plastically deformed HPHT-treated alloy at 35% tensile strain, showing the obvious dislocations and stacking faults (SFs) in the FCC and B2 phases. The insets show the related selected area electron diffraction (SAED) patterns of FCC and B2 phases (indicated by yellow and blue colors, respectively). **f** Generation of deformation twins in the FCC phase (the region highlighted by the red dashed circle in **e**). The inset shows the corresponding fast Fourier transform

(FFT) pattern. **g** High-resolution TEM image of the interface between FCC and BCC phases shown in **e**. **h** Inverse fast Fourier transform (IFFT) micrograph for the TEM image in **g**, where dislocations are highlighted by the yellow dashed circles. **i** High-resolution TEM image for the BCC phase in **e**. Geometrical phase analysis (GPA) image over the white dashed region illustrating the epsilon_xy strain mapping in the BCC phase. **j** Schematic sketch of microstructure evolution with increased plastic strain in our HPHT-treated alloy. The HPHT-treated alloy with a hierarchically patterned microstructure exhibits distinctive multiple deformation mechanisms, including dislocations, stacking faults, microbands and deformation twins. The coherent interface holds a key to the activation of the multiple deformation mechanisms, as shown in the enlarged sketch maps.

Formation of extended stacking faults (SFs) near the interfaces is also identified, which has been regarded as a mechanism that accommodates local interfacial stresses[41]. With strain increasing to 15%, SFs are activated in the FCC phase with a much higher density as compared with those at 5% strain, indicating enhanced dislocation activities in the FCC phase. The accumulated dislocation storage has led to the formation of microbands, which pile up and evolve into high-density dislocation walls (HDDWs)[42] near the phase boundary, further contributing to strain hardening capability of the alloy[43]. In the B2 phase, dislocations and slip traces are widely observed (Supplementary Fig. 13), in sharp contrast to the nearly unnoticeable dislocation activities in the Ho alloy (Supplementary Fig. 17). As the strain increases to 35%, numerous microbands intersect each other, and dense dislocation networks form in the FCC regions. Some microbands extend from the FCC phase and terminate at the B2 phase, while some microbands cut across the B2 phase, which is important for the strain hardening and uniform deformation of the HPHT alloy. More interestingly, we observed clear traits of deformation twins (DTs) in the highly deformed hierarchically patterned structure (indicated by red arrows in Fig. 4d). An enlarged view shows a pattern of intensely intersected DTs and microbands (inset in Fig. 4d). Note that neither

HDDWs nor DTs were observed in the Ho alloy at fracture. Such progressive activation of multiple plastic deformation mechanisms in both FCC and BCC phases of the HPHT alloy is in distinct contrast to the case of Ho alloy, where only limited slip lines exist in the FCC phase while the BCC phase exhibits no evident dislocation activities at large strains.

The TEM image of the fracture region captured at 35% strain (Fig. 4e) also demonstrates slip transfer between the FCC and B2 phases (confirmed with the SAED pattern in the insets) and numerous dislocation-SF interactions in the HPHT-treated alloy, different from the localized deformation at the interface and the absence of dislocation in the B2 phase in the Ho alloy (Supplementary Figs. 17 and 19). In the FCC phase, emanating deformation twins are occasionally observed in regions filled with tangled dislocations and terminate in the B2 phase (red dashed circle in Fig. 4e). A magnified twin structure is shown in Fig. 4f, with the inset showing the corresponding SAED pattern. These synergistic deformation mechanisms activated in the HPHT-treated alloy have led to a less coherent FCC/B2 interface after fracture (see the HRTEM image in Fig. 4g), where dislocations are frequently identified on both sides of the interface (highlighted with yellow dashed circles in Fig. 4h). For comparison, the inverse fast

Fourier transform (IFFT) micrograph of the HRTEM image captured for the Ho alloy (Supplementary Fig. 17) only contains edge dislocations in the FCC phase. These results indicate that the uniform deformation of the HPHT-treated alloy is strongly related to the compatible deformation of the B2 phase, which is found to be associated with high-density dislocations after fracture (confirmed by the geometrical phase analysis in Fig. 4i).

The microstructure evolution of the HPHT-treated alloy has been summarized in Fig. 4j. Dislocations preferentially initiate in the softer FCC phase, and subsequently are allowed to transfer into the harder B2 phase due to the highly coherent interface, which effectively alleviates stress concentration at the interface and coordinates deformation between the two phases. With increasing strain, dislocation density increases in both phases, leading to the formation of microbands and high-density dislocation walls (HDDWs) near the interface, contributing to strain-hardening[44]. The microbands can also transmit across the interface, further facilitating strain delocalization[45]. At a later stage of deformation, as the coherency of the FCC/B2 interfaces decays, deformation twins are activated to further accommodate deformation, subdividing microbands and promoting mutual interactions between refined microbands. Meanwhile, the less coherent interfaces can act as strong barriers to dislocation motion[46] and pin the movement of refined microbands in both phases. Overall, the hierarchically patterned structure with the coherent interface (Fig. 1h–l) not only confines dislocation slip, but more importantly, it also provides a route to accommodate plasticity through the activation of multiple deformation mechanisms, including SFs, microbands and deformation twins (illustrated in Fig. 4j), thus endowing the alloy with excellent structural stability and strain-hardening capability.

In summary, we have demonstrated that HPHT treatment, typically adopted to study earth materials and synthesize superhard materials in industry, can be used to overcome the trade-off between strength and ductility in a eutectic high entropy alloy. The HPHT-treated alloy exhibits an exceptional combination of high strength and large uniform elongation without encountering interface-induced embrittlement and plastic instability frequently observed in eutectic high entropy alloys. The HPHT treatment modifies the microstructure and the interface structure, both playing a critical role in mediating strain delocalization and activating multiple deformation mechanisms. In view of the adjustable pressure, temperature, and chamber size of the self-designed hexahedron-anvil-press apparatus, the HPHT strategy can be generally applied to pure metal copper (Supplementary Fig. 22) and other alloy systems (Supplementary Fig. 23 and Supplementary Fig. 24), making it an appealing pathway to the design and fabrication of advanced structural materials with extraordinary mechanical properties.

## Methods

### Materials

Alloy with a nominal composition of $Al_{0.7}CoCrFeNi$ (0.7 in molar ratio) was prepared by arc-melting with a purity higher than 99.95 (wt.%). The alloy was re-melted five times to promote homogeneity in a Ti-gettered high-purity argon atmosphere. The melted alloy was cast into a water-cooled copper mould to form a plate with a length of 85 mm, a width of 11 mm, and a thickness of 5 mm. The obtained as-cast alloy was denoted as the Ac alloy. To reduce micro-segregations from the casting process, the Ac alloy was homogenized at 1423 K for 8 h and then water-quenched to ambient temperature to obtain the Ho alloy.

### HPHT treatment

A multi-anvil apparatus was used for high-pressure and high temperature experiments. The schematic diagram of the high-pressure device in a large volume cubic press is shown in Fig. 1f. Six WC anvils are fixed on six pistons and simultaneously pushed by six hydraulic cylinders to operate the cubic press. The loading force acts on the six outer surfaces of the cubic assembly from six directions, squeezing the pyrophyllite pressure transmitting medium and establishing high pressure inside the chamber. The temperature was controlled by changing electrical power exerted on the graphite tube heater and measured by Type C W-Re thermocouples[47]. The pressure was determined from previously determined calibration curves. The Ho alloy sample with a diameter of 10 mm and a thickness of 5 mm was placed in the middle of the cubic assembly. After the cell was thoroughly clamped, the sample was subjected to a pressure of 6 GPa, heated up to 1473 K, and then maintained under the target pressure and temperature for 2 h. After that, the sample was cooled down to room temperature and the pressure was then released. Finally, the recovered HPHT-treated sample was removed from the pressure chamber. To study the temperature effect on the mechanical properties of the HPHT-treated EHEA, additional Ho alloy samples were treated at 1373 K and 1573 K for 2 h, which are denoted as HPHT-treated-1373K and HPHT-treated-1573K, respectively. Among these samples, the HPHT-treated alloy (treated at 1473 K) exhibits the optimal mechanical property, as shown in the main text. The HPHT-treated-1373K and the HPHT-treated-1573K samples were mainly used for comparison (see Supplementary Note 2, and Supplementary Fig. 9 for more details).

### Mechanical property testing

Flat, dog-bone-shaped tensile samples with a gauge length of 4 mm, a width of 2 mm and a thickness of 1 mm were extracted from the HPHT-treated alloys by using electrical discharge machining. Before testing, the sides of the gauge section were carefully ground using SiC grinding papers. Uniaxial quasi-static tensile tests were conducted using a Zeptool S-5000 machine at a strain rate of $3 \times 10^{-4}$ s$^{-1}$ at room temperature. The strain was measured by a non-contact Linconst video extensometer. The tests were repeated two to three times for each type of sample. Fracture surfaces of the specimens after tensile tests were examined by a field-emission scanning electron microscope.

### Microstructure characterization

EBSD and ECCI characterization were carried out by using a JEOL JSM-7900F instrument. The chemical homogeneity was studied using EDS. The specimens for EBSD and ECCI were prepared by electro-polishing in a solution of $HNO_3$ (20%) and $C_2H_5OH$ (80%) with a voltage of 20 V at 233 K. EBSD measurements were carried out at 15 kV. The average GND density can be estimated by using the strain gradient theory[48]: $\rho_{GND} = 2\theta/ub$, where $\rho_{GND}$ is the GND density at the point of interest, $\theta$ is the local misorientation, b is the Burger's vector calculated from the XRD pattern and u is the unit length (step size = 0.1 μm).

The ECCI technique was conducted at 30 kV using a backscattered electron (BSE) detector in the SEM (JSM-7900F). The X-ray diffraction was performed using Cu Kα radiation, and the diffraction patterns were recorded over a 2θ range from 20° to 100° with a scanning speed of 4°/min. The specimens for TEM were prepared using a focused ion beam (FIB, Thermo Scientific Scios 2) technique. The TEM analysis was conducted using an FEI Tecnai F30 operated at 300 kV fitted with a STEM-EDS detector. According to the Bramfitt lattice mismatch theory[49], the mismatch factor δ of FCC and BCC phase is calculated as: $\delta = |d_{[uvw]m} - d_{[uvw]s}|/d_{[uvw]}$, where $d_{[uvw]}$ is the interatomic spacing along the direction with miller index of uvw; subscripts m and s represent the matrix phase and the second phase, respectively.

## Data availability

The data that support the findings of this study are available from the corresponding authors on request.

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

## Acknowledgements

Y.T. acknowledges financial support from the National Natural Science Foundation of China (12202381) and China Postdoctoral Science Foundation (2022M712758). H.Z. acknowledges financial support from the National Natural Science Foundation of China (12222210, 12172324) and Zhejiang University K. P. Chao's High Technology Development Foundation. H.W. acknowledges financial support from the National Natural Science Foundation of China (52073254). H.G. acknowledges a start-up grant from Nanyang Technological University and Agency for Science, Technology and Research (A*STAR). We thank N.R. Kwesi, X. Wu, Z.L. Zhang, G.G. Tang and Y.C. Wu for their experimental assistance. We thank L. Lu for discussions.

## Author contributions

Y.T. and H.F.Z. proposed the idea. H.F.Z., H.J.G. and H.K.W. directed the project. Y.T., H.F.Z. and H.K.W. designed the experimental programme. Y.T. carried out the main experiments. H.K.W., Y.T., C.W., Z.Q.H., J.K.W., Z.C.Z., H.L. and Y.K.Y. constructed the multi-anvil apparatus for high-pressure and high temperature experiments. Y.T. and C.W. performed the SEM microstructure characterization and mechanical testing. Y.T. analyzed EBSD patterns for the calculation of GND. Y.T. and X.C.L. conducted the ECCI characterization and analyzed the data. Y.T. and H.F.Z. performed the TEM characterization and analyses. Y.T., H.F.Z., H.J.G., H.K.W., X.P.O.Y., Q.S.H., Q.K.Z., Q.Z. and W.Y. discussed the results. Y.T., H.F.Z. and H.J.G. wrote the manuscript. All co-authors contributed to the final manuscript.

## Competing interests

The authors declare no competing interests.
