## [Peer Review File · Nature Communications]

Overcoming strength-ductility tradeoff with high pressure thermal treatmentREVIEWER COMMENTS

Reviewer #1 (Remarks to the Author):

In this manuscript, the Authors have suggested the combination of high-pressure and high-temperature treatments to get both high strength and relatively high ductility in structural materials. This is a nice work, presenting very significant results from a scientific point of view. The formation of a hierarchically patterned microstructure with coherent interfaces, which can promote multiple deformation mechanisms, seems to be a good concept overcoming the strength-ductility tradeoff. Considering both the unique, good experimental results and their scientific significance, I would recommend the publication of this manuscript in Nature Communications, after minor corrections, according to the following comments:

- 1) When quantifying the plastic deformation accommodation in FCC and BCC (B2) phases, the Authors must explain – shortly - why only the GND dislocation density should be taken into account, rather than the total one. Furthermore, the Authors should check the formula given in Methods, cited from Ref. (48) for estimating the GND density, since such a formula is not found in Ref. (48) (in the work of Gao et al., Journal of the Mechanics and Physics of Solids, 1999)
- 2) The authors have showed the unique increasing effect of HPHT treatment on the strength and ductility of Ho samples. The results shown in Figure 2a also show that the HPHT treatment rather reduced the strength of the as-cst(Ac) samples. Why?
- 3) It is completely appropriate to use engineering strain to view the improvement of ductility. It is also completely appropriate to use the engineering stress in the case of small deformation. In the case of large tensile deformation (elongation), however, the cross-section of the sample can be significantly reduced, so it is advisable to present the true stress. In the present case, with 35% tensile deformation, the true stress may be 30% higher than the engineering value. Representing the true stress, the ultimate tensile stress of the Ho+HPHT sample (the data point "This work" in Fig. 2b) would be more than 1500 MPa.

Reviewer #2 (Remarks to the Author):

In this study, the authors suggested a novel thermo-mechanical process for enhancing mechanical properties of eutectic HEA. Although the paper dealt with interesting topic, the scientific novelty is not enough to publish this paper in Nature Communications. Therefore, it is difficult to recommend the acceptance of this paper. Detailed comments are listed as below:

1. In this paper, the authors suggested that high-pressure and high-temperature (HPHT) treatment is novel process for enhancing the mechanical properties of the metallic materials. However, the HPHT treatment is quite similar process with hot isotatic pressing (HIP) treatment, which has been conventionally used for controlling microstructure and properties of the materials.
2. The title of this paper is "Overcoming strength-ductility tradeoff with high pressure thermal treatment". This implies that the process can be used to improve the mechanical properties of most metal materials. However, the reviewer pointed out that the improvement of strength-ductility combination by HPHT treatment is attributed to microstructure refinement of only eutectic HEA, not general alloys. It demonstrates that the process-induced strengthening in the present study can be only driven in the materials having specific microstructure characteristics (i.e., eutectic HEAs). Although the authors shows the effect of HPHT on microstructure and mechanical properties of pure copper in Supplementary materials, the HPHT-treated sample exhibit only strength improvement (by grain refinement), not ductility improvement as compared to as-homogenized sample.
3. For the Al-containing CoCrFeNi HEAs, numerous studies have been performed with respect to microstructure control to enhance strength-ductility combination via grain refinement or hetero-structuring for past decade.
4. In order to emphasize that the HPHT process is superior for enhancing mechanical of the eutectic

HEAs, the authors should investigate the process-induced microstructure change in not only Al_{0.7}CoCrFeNi HEA but also AlCoCrFeNi_{2.1}, the most famous eutectic HEA [Nat. Comm. 10 (2019) 489, Nature 608 (2022) 62-68], or other eutectic HEAs. This will strongly support the versatility and usefulness of this process for improving properties for the eutectic HEA.

5. It has been well known that the mechanical properties of the as-cast or as-homogenized ingot are lower than that of the samples treated by thermo-mechanical process (i.e., forging, heat treatment) due to coarse grain structure of the ingots. The authors should show the tensile curves of not only as-homogenized ingot but also the sample treated by conventional thermo-mechanical process for emphasizing the properties of the sample treated by HPHT.

Minor comments.

1. The global chemical composition of the alloy should be presented. Moreover, the chemical composition of each phase (i.e., FCC and B2) should be presented.

2. The regions at different local strains for EBSD analysis of the deformed sample should be indicated for the readers.

3. In order to investigate GND evolution in the sample during deformation, the EBSD analysis in higher magnification is needed for higher accuracy. Moreover, the authors should present EBSD GND maps, not KAM maps.

4. As shown in Fig. 2(a), the strain hardening ability of the HPHTed sample is similar with that of as-homogenized sample. Moreover, it seems that the strain hardening rate of Ac+HPHTed sample is higher than Ho-HPHTed sample. The strain hardening rate is highly related to strength-ductility combination of metallic materials. The authors should discuss the strain hardening behaviors of the samples.

Reviewer #3 (Remarks to the Author):

Developing unique structures with substantial improvement of both strength and ductility is highly challenging. Heterostructures materials have been recently reintroduced, among them so-called harmonic structures. However, such a methodology has yet to be utilized on high entropy alloys, especially on eutectic HEAs. Indeed, the absence of characterization of these alloys' more or less fragile behavior has already slowed down attempts to develop HEA alloys as structural materials. From this point of view, the study presented here is an approach that could benefit the community. The analysis is well-detailed, and the process engineering approach is attractive, clear, and easily reproduced by the interested community. Below are some comments that the authors could comment on/discuss:

1. This study lacks statistics and quantification, which raises the question of whether the observed effects and the underlying mechanisms are anecdotal or local (regarding observations at very fine scales). The absence of this analysis would remove all the interests of the discussion.

2. A rather critical point concerns using KAM to estimate the density of GNDs. Regarding the EBSD data, it is trendy to use an equation immediately if it has been used somewhere in a paper. But while KAM is essentially a scalar number, the sought GND density is tensor and should be determined as such. So, computing dislocations density via KAM is generally wrong, and this becomes obvious in the case of an elastically bent crystal domain with no dislocations. Indeed, the proposed treatment of EBSD data will give a senseless non-zero result here. A few interesting investigations from EDAX-TSL remarkably showed that a different step size results in different GNDs. How trustful can GNDs be if it depends on the acquisition conditions? This has a significant impact on the whole discussion that follows. Notice that techniques such as Convolutional Multiple Whole Profile fitting of DRX patterns are more efficient and robust.

3. It is said that different properties can be generated by varying the experimental conditions. The supplementary Fig. 5, which is supposed to reflect this assertion, must be more conclusive. Without any quantification, the two figures are very close regarding mechanical characteristics and appearance. Did I miss something?

Point-by-point Response to Review Comments

RE: NCOMMS-23-46785

Title: Overcoming strength-ductility tradeoff with high pressure thermal treatment

We highly appreciate the reviewers' constructive comments and valuable suggestions on our manuscript. Based on these comments, we have carefully revised the manuscript. In the following, the review comments are listed in *italic* blue font and our response to each comment is given in **black** font.

Reviewer #1

In this manuscript, the Authors have suggested the combination of high-pressure and high-temperature treatments to get both high strength and relatively high ductility in structural materials. This is a nice work, presenting very significant results from a scientific point of view. The formation of a hierarchically patterned microstructure with coherent interfaces, which can promote multiple deformation mechanisms, seems to be a good concept overcoming the strength-ductility tradeoff. Considering both the unique, good experimental results and their scientific significance, I would recommend the publication of this manuscript in Nature Communications, after minor corrections, according to the following comments.

Reply: We sincerely thank the reviewer for giving positive comments on the importance and novelty of our work.

1) When quantifying the plastic deformation accommodation in FCC and BCC (B2) phases, the Authors must explain - shortly - why only the GND dislocation density should be taken into account, rather than the total one. Furthermore, the Authors should check the formula given in Methods, cited from Ref. (48) for estimating the GND density, since such a formula is not found in Ref. (48) (in the work of Gao et al., Journal of the Mechanics and Physics of Solids, 1999)

Reply: We deeply appreciate the reviewer's comments. As pointed out by the reviewer, to quantify the plastic deformation accommodation in FCC and BCC phases, we calculated the average densities of geometrically necessary dislocations (GNDs) under various strains in the HPHT-treated alloy using EBSD kernel average misorientation (KAM) analysis. The KAM value directly reveals deformation-induced local orientation gradients inside the grains. Accommodation of such strain gradients needs the storage of geometrically necessary dislocations (GNDs). Thus, the KAM value is associated with the density of geometrically necessary dislocations (GNDs) in the detected region,

higher KAM values suggest a higher density of GNDs and more plastic strains in these zones. In fact, many studies have used the measured KAM (D. Raabe et al., Nature 2022;608;301-316. Y.M. Wang et al., Nat. Mater. 2018;17;63-71) and the estimated GND (S.B. Gao et al., Nat. Commun. 2023;14;6752. Z.M. Li et al., Acta Mater. 2017;131;323-335) to show the distributions of deformation-induced misorientations. We have measured identical values of GND density in both phases at fracture strain, suggesting homogeneous plastic deformation distribution in the HPHT-treated alloy.

Following the reviewer's valuable suggestion, we have now performed additional experiments to take into account the total dislocation density. The total dislocation density can be estimated from the X-ray diffraction (XRD) patterns by employing the Williamson-Hall method (G.K. Williamson, W.H. Hall, Acta Metall. Mater. 1953;1;22-31. D. Raabe et al. Nature 2022;608;301-316). The XRD patterns of the tensile-deformed Ho and HPHT-treated alloy at various strain amplitudes are presented in Fig. R1.

Figure R1 The X-ray diffraction patterns of the Ho and HPHT-treated alloys at different strain amplitudes.

Figure R2 a, Dislocation density against strain in the FCC and the B2 phases for the Ho alloy. **b**, Dislocation density against strain in the FCC and the B2 phases for the HPHT-treated alloy.

The calculated dislocation densities are illustrated in Fig. R2. As shown in Fig. R2a, of the Ho alloy increases from $2.68 \times 10^{14} \text{ m}^{-2}$ to $3.75 \times 10^{14} \text{ m}^{-2}$ from 0% to 3% strains, whereas the

increases from $7.51 \times 10^{13} \text{ m}^{-2}$ to $1.81 \times 10^{14} \text{ m}^{-2}$. Upon further loading, the dislocation density of the FCC phase increases to $9.79 \times 10^{14} \text{ m}^{-2}$ at 15% strain, which is higher than that of the B2 phase. In the HPHT-treated alloy (Fig. R2b), the dislocation density of the FCC phase increases rapidly from $2.86 \times 10^{14} \text{ m}^{-2}$ to $6.99 \times 10^{14} \text{ m}^{-2}$ from 0% to 15% strains, whereas the dislocation density of the B2 phase increases from $1.03 \times 10^{14} \text{ m}^{-2}$ to $1.04 \times 10^{15} \text{ m}^{-2}$. Upon further loading, the dislocation density of the B2 phase increases to $1.91 \times 10^{15} \text{ m}^{-2}$ at 35% strain, whereas the dislocation density of the FCC phase increases to $1.63 \times 10^{15} \text{ m}^{-2}$ at 35% strain. These results confirm the uniform deformation of the HPHT-treated alloy. In addition, the dislocation densities in the HPHT alloy are much higher than those in the Ho alloy, suggesting improved capability of dislocation storage in the HPHT-treated alloy. We have now added additional discussion and new results in the revised Supplementary Information to support the difference in plastic deformation accommodation behaviors in Ho alloy and HPHT-treated alloy.

In addition, we deeply appreciate the reviewer for the careful review that points out the citation issue with Ref. (48). Based on the strain gradient model in Ref. (48) (Gao et al., J. Mech. Phys. Solids 1999;47;1239-1263), Kubin and Mortensen defined the GND calculation method (L.P. Kubin, A. Mortensen, Scr. Mater. 2003;48;119-125). In the revised manuscript, we have now corrected Ref. (48).

2) The authors have showed the unique increasing effect of HPHT treatment on the strength and ductility of Ho samples. The results shown in Figure 2a also show that the HPHT treatment rather reduced the strength of the as-cast(Ac) samples. Why?

Reply: We sincerely thank the reviewer for this valuable comment. The reduction in the strength of the as-cast(Ac) samples can be attributed to the change in the phase distribution. As shown in Fig. R3, the Ac sample exhibits a coarse columnar structure and significant B2 phase segregation. The agglomerated B2 region has limited the plastic deformability, which can be an obstacle to the strengthening of the Ac sample. At the same time, the B2 phase has no tensile ductility, due to the instability of the B2 phase to accommodate shape changes (J. Joseph et al., J. Alloys Compd. 2017;726;885-895). For the Ac sample, the HPHT treatment involves annealing and destressing processes. With HPHT treatment, the EBSD map reveals a uniform distribution of the B2 phase, without significant phase segregation. Thus, the plastic deformation ability of the B2 phase can be tuned. These changes in the microstructure contribute to the improvement of tensile ductility in the HPHT-treated alloy and the reduction of strength, similar to other HIP-processed alloys in the literature (M.T. Tran et al., Mater. Sci. Eng. A 2021;828;142110). In the revised manuscript, we have added a few sentences to explain this point.

Figure R3 a, An electron backscatter diffraction (EBSD) inverse-pole figure (IPF) map of the Ac $\text{Al}_{0.7}\text{COCrFeNi}$ alloy. **b**, The corresponding EBSD phase map reveals that the dual-phase lamellae consist of alternating FCC layers and B2 layers in the Ac alloy. **c**, The electron backscatter diffraction (EBSD) inverse-pole figure (IPF) map of the Ac+HPHT $\text{Al}_{0.7}\text{COCrFeNi}$ alloy. **d**, The corresponding EBSD phase map reveals that the dual-phase lamellae structure has been modified in the Ac+HPHT alloy.

3) It is completely appropriate to use engineering strain to view the improvement of ductility. It is also completely appropriate to use the engineering stress in the case of small deformation. In the case of large tensile deformation (elongation), however, the cross-section of the sample can be significantly reduced, so it is advisable to present the true stress. In the present case, with 35% tensile deformation, the true stress may be 30% higher than the engineering value. Representing the true stress, the ultimate tensile stress of the Ho+HPHT sample (the data point "This work" in Fig. 2b) would be more than 1500 MPa.

Reply: We thank the reviewer for this suggestion. We fully agree with the reviewer that it is necessary to present the true stress-strain curve. As shown in Fig. R4, the tensile true stress-strain curves of the Ac, Ac+HPHT, Ho and Ho+HPHT alloys exhibit remarkable differences in fracture stress and strain. The Ac+HPHT alloy exhibits a remarkable improvement in fracture true strain compared to the Ac alloy, while the fracture true strength is almost the same. The Ho alloy only has a fracture true strain of ~14.7%. The HPHT alloy doubles the fracture strain to ~30.7%. More importantly, the HPHT alloy exhibits a high fracture true stress of ~1570 MPa, which is much

higher than those of the AC, Ac+HPHT and Ho alloys, respectively. We have added the true stress-true strain curves in the Supplementary Information.

Figure R4 True stress-strain curves of the homogenized alloy (Ho alloy) after HPHT treatment, along with those of the Ho alloy before HPHT treatment and the as-cast alloy (Ac alloy) with and without HPHT treatment.

Reviewer #2

In this study, the authors suggested a novel thermo-mechanical process for enhancing mechanical properties of eutectic HEA. Although the paper dealt with interesting topic, the scientific novelty is not enough to publish this paper in Nature Communications. Therefore, it is difficult to recommend the acceptance of this paper. Detailed comments are listed as below:

Reply: We thank the reviewer for the positive comment on the topic of our manuscript. Before point-by-point responding to the comments, we would like to restate the novelty of our work in the following aspects.

First, we have achieved synergistic enhancement in strength and ductility of structural materials by a combination of hydrostatic pressure and temperature for the first time. Given the vast space of control parameters involved in the HPHT treatment and the often complex microstructural evolution under extreme pressures and temperatures (A.P. Zhilyaev et al., Prog. Mater. Sci. 2008;53;893-979. W.Q. Wang et al., Metals 2019;9;867), it is extremely challenging to predict and interpret the mechanical behaviors of HPHT-treated structural materials (X. Sauvage et al., Mater. Sci. Eng. A 2012;540;1-12. X. Zhou Nature 2020;579;67-72). The understanding of process-microstructure-mechanical properties relationship with HPHT is thus of scientific importance. By using the ‘large-volume’ press technology (Fig. 1f in the manuscript) and identifying viable pressure-temperature pathways, we demonstrate that an initially strong-yet-brittle eutectic high entropy alloy exhibits an exceptional combination of high strength and large uniform elongation after the HPHT treatment (Fig. 2 in the manuscript), without encountering any interface-induced embrittlement and plastic instability frequently observed in traditional eutectic high entropy alloys (Q. Wu et al., Nat. Commun. 2022;13;4697).

Moreover, we would like to point out that the HPHT technology (R.C. Liebermann et al., High Pressure Res. 2011;31;493-532) is very different from the traditional HIP (M.H. Bocanegra-Bernal, J. Mater. Sci. 2017;726;885-895) or HPT technology (A.P. Zhilyaev et al., Prog. Mater. Sci. 2008;53;893-979), which will be discussed in details below. Processing conditions of high pressure and high temperature are challenging, especially for the experimentally high-pressure treatment of large-volume bulk materials. Therefore, we believe that our strategy is of considerable technical interests to the community. Our results have demonstrated the HPHT treatment has a great potential for altering the microstructure and synergistically improving mechanical properties in structural materials.

Second, we have revealed the formation of a hierarchically patterned BCC/FCC microstructure with coherent interfaces by HPHT processing, and the multiscale deformation mechanisms underlying its strength-ductility synergy is novel. Our Electron contrast channel imaging (ECCI) analysis and transmission electron microscopy (TEM) results for the HPHT-treated

alloy have demonstrated that the coherent interfaces promote the synergic deformation of the hard and soft phases, which leads to the activation of multiple deformation mechanisms, including dislocations, stacking faults, microbands and deformation twins, at multiple length scales (Fig. 4 in the manuscript). Such progressive activation of multiple plastic deformation mechanisms in the HPHT alloy is in sharp contrast to the case of alloy without HPHT treatment, where only limited slip lines exist in the FCC phase while the BCC phase exhibits no evident dislocation activities at large strains (Fig. 10 in the supplementary information). More importantly, isolated microcracks are uniformly distributed near the fracture surface in the HPHT-treated alloys, in contrast to the catastrophic fracture in those without HPHT treatment (Fig. 6 in the supplementary information). The unique HPHT-induced hierarchically patterned microstructure with coherent interfaces helps relieve stress concentration at the interfaces, thereby arresting interfacial cracking commonly observed in traditional eutectic high entropy alloys. We believe that our findings offer a promising paradigm for tailoring microstructure and mechanical properties with the combination of pressure and temperature and advance our fundamental understanding of the intrinsic deformation behavior of HPHT-treated alloys.

Last but not least, the HPHT strategy can be generally applied to other metallic materials (including those suggested in your valuable comments) for synergistic improvement of mechanical properties (Figs. R7, R8 and R9), which will be shown in details below. In comparison with conventional processing approaches involving high pressure conditions, such as high-pressure torsion and equal-channel angular pressing, HPHT treatment is attractive in that it is non-destructive and free of severe plastic deformation during processing. It is also a cost-effective processing route for fabrication, providing opportunities to develop unique microstructures and mechanical properties which are not easily attained by conventional processing approaches. HPHT treatment also permits the processing of relatively large bulk materials (the diameter of the largest sample in our apparatus reaches 80 mm), rendering it appropriate for industrial implementation.

Overall, we believe that our work have provided a new and promising research direction for the design of bulk metallic materials with optimal properties. Our point-by-point responses to your constructive comments have been listed in details below.

1) In this paper, the authors suggested that high-pressure and high-temperature (HPHT) treatment is novel process for enhancing the mechanical properties of the metallic materials. However, the HPHT treatment is quite similar process with hot isotatic pressing (HIP) treatment, which has been conventionally used for controlling microstructrue and properties of the materials.

Reply: We thank the reviewer for this comment. We would like to point out that the high-pressure and high-temperature (HPHT) treatment reported in this work is quite different from the

conventional hot isostatic pressing (HIP) treatment. To help address this comment, we have plotted a comparison of the HIP treatment and HPHT treatment in Fig. R5.

The hot isostatic pressing (HIP) treatment has been used for upgrading castings, densifying presintered components, consolidating powders, and interfacial bonding (M.H. Bocanegra-bernal, *J. Mater. Sci.* 2017;726;885-895). It involves the simultaneous application of pressure and temperature in a specially constructed vessel. The pressure is applied with a gas. The pressure of the HIP treatment is usually about **100 MPa** (R. Yang et al., *Intermetallics* 2023;159;107929. H. Vashishtha et al., *Mater. Charact.* 2023;205;113304. M.T. Tran et al., *Mater. Sci. Eng. A* 2021;828;142110. W.D. Xuan et al., *J. Mater. Res. Technol.* 2021;14;1609-1617. J.W. Wang et al., *Adv. Mater. Res.* 2012;472;696-699. H. Huang et al., *J. Mater. Eng. Perform.* 1998;7;784-788), which is much lower than that of our HPHT treatment (**6 GPa**). In HIP treatment, although the pressure is isostatic, shrinkage is not generally isotropic, particularly if containment is used (H.V. Atkinson et al., *Metall. Mater. Trans. A* 2000;31;2982). There is a problem with using gas as the pressure medium for the HIP treatment. The reduction in pore surface energy takes place during the HIP process and acts as a driving force for pore closure, however, the entrapped gas backpressure inside the pore reacts adversely (A. du Plessis et al., *Addit. Manuf.* 2020;34;101191). Once a pore starts to shrink through the HIP, the entrapped gas pressure boosts up inside the pores and prohibits further closure. Therefore, the closure of inner pores was correlated with the trapped gas, which could not completely dissolve during the HIP treatment and then remained pores with high internal pressures. Therefore, the pores with gas inside were hardly closed during HIP. To avoid the above problems in HPHT treatment, six WC anvils are fixed on six pistons and simultaneously pushed by six hydraulic cylinders to operate the cubic press. The six anvils define a center cubic cavity, inside of which is the cubic assembly in our experiments, as shown in Fig. R5b. The motion of the six anvils compresses the cubic assembly so that the sample chamber pressure continues to build up in the assembly with a decrease in the volume of the pyrophyllite cube. Such a combination of pressure transmission media can produce a quasi-hydrostatic stress state and obtain a fully densified sample.

Figure R5 a, Schematic diagram of the HIP treatment. **b**, Schematic diagram of the HPHT treatment set-up and the sample assembly part.

Moreover, grain coarsening is a common phenomenon during the HIP process, having an unfavorable effect on mechanical properties that need to be considered (X. Yang et al., *Mater. Lett.* 2022;309;131334. A. Rezaei et al., *Mater. Sci. Eng. A* 2021;823;141721. M.T. Tran et al., *Mater. Sci. Eng. A* 2021;828;142110). Because of the commonly slow cooling rate associated with HIP treatment, the harmful phases may precipitate during the cooling step. For example, Joseph et al. observed the formation of σ phase in an $\text{Al}_{0.85}\text{CoCrFeNi}$ alloy produced by HIP treatment, such microstructural evolution resulted in a sharp loss in ductility (Joseph J et al., *Mater Sci Eng* 2018;733;59-70). Remarkably, the HPHT-treated alloy in our work demonstrated exceptional grain coarsening resistance with negligible grain growth. At the same time, the high solidification rates associated with the HPHT fabrication route do not allow sufficient time for the harmful phases to form.

Figure R6 True stress-strain curves of the homogenized alloy (Ho alloy) after HPHT treatment (Ho+HIP), along with the Ho alloy before HIP treatment.

Most importantly, the effect of HIP treatment on the microstructure and mechanical behavior of HEAs is no different from that of traditional processing methods. For example, the HIP-processing of the $\text{Al}_{0.85}\text{CoCrFeNi}$ alloy resulted in coarsening of the microstructure and resulted in the significant loss of ductility after processing by HIP (Joseph J et al., *Mater Sci Eng* 2018;733;59-70). At the same time, although the HIP can improve the ductility of high entropy alloy, it unfortunately reduces the strength (M.T. Tran et al., *Mater. Sci. Eng. A* 2021;828;142110). These results suggest that HIP treatment has limitations in overcoming the strength-ductility trade-off of structural materials. To demonstrate the effect of HIP treatment on the mechanical property of the $\text{Al}_{0.7}\text{CoCrFeNi}$ alloy, we have performed additional experiments with hot isostatic pressing at 1473K under 150 MPa for a holding time of 2 hours and then uniaxial quasi-static tensile tested to evaluate the mechanical property. Fig. R6 shows the engineering stress-strain curves of the $\text{Al}_{0.7}\text{CoCrFeNi}$ alloy in both the homogenized and HIP-treated conditions (Ho+HIP) for tensile testing at room temperature. A noticeable increase in tensile strength occurs after the HIP treatment,

the elongation decreases a little. In contrast, our HPHT-treated alloy exhibits nearly 100% enhancement in both strength and tensile ductility compared with the Ho alloy. Such unusual HPHT-induced strength-ductility synergy is attributed to the formation of a hierarchically patterned BCC/FCC microstructure with coherent interfaces, which promotes the activation of multiple deformation mechanisms at multiple length scales.

Overall, the proposed HPHT treatment is fundamentally different from the previously reported HIP treatment in terms of the processing method, adjustment of microstructure and impact on macroscopic mechanical properties. The HPHT treatment provides a novel and promising route for tailoring the mechanical properties of structural materials. In the revised manuscript, following the reviewer's suggestion, we have added the additional results and discussions in the Supplementary Information to explain the difference between HPHT treatment and HIP treatment.

2) The title of this paper is “Overcoming strength-ductility tradeoff with high pressure thermal treatment”. This implies that the process can be used to improve the mechanical properties of most metal materials. However, the reviewer pointed out that the improvement of strength-ductility combination by HPHT treatment is attributed to microstructure refinement of only eutectic HEA, not general alloys. It demonstrates that the process-induced strengthening in the present study can be only driven in the materials having specific microstructure characteristics (i.e., eutectic HEAs). Although the authors show the effect of HPHT on microstructure and mechanical properties of pure copper in Supplementary materials, the HPHT-treated sample exhibit only strength improvement (by grain refinement), not ductility improvement as compared to as-homogenized sample.

Reply: We thank the reviewer for the comment. Indeed, we have originally applied the high-pressure and high-temperature (HPHT) treatment strategy on eutectic high entropy alloys to help overcome the strength-ductility trade-off. Compared with conventional metallic materials, eutectic high entropy alloys represent a typical class of multi-principal-element alloys featured by a hierarchical microstructure of dual-phase lamellae colonies, which offers great potential for achieving superior mechanical properties. Similar to conventional metallic materials, pronounced dislocation pile-ups are routinely developed in the vicinity of internal boundaries during mechanical straining for the eutectic high entropy alloys, which leads to large local stress concentration and accelerates the onset of micro-cracking along the internal boundaries and catastrophic damage. Using a brittle eutectic high entropy alloy, we have demonstrated that our HPHT strategy can help overcome the long-standing dilemma by suppressing the typical brittle interfacial fracture.

To verify that our HPHT strategy can also be implemented for other metallic materials, we have now performed the same heat treatment engineering protocol to realize the process-induced

strengthening in a $\text{Fe}_{50}\text{Mn}_{27}\text{Ni}_{10}\text{Cr}_{13}$ alloy system. Fig. R7 displays the tensile properties of the Ho and HPHT-treated $\text{Fe}_{50}\text{Mn}_{27}\text{Ni}_{10}\text{Cr}_{13}$ samples at room temperature. The HPHT-treated sample exhibits greatly improved fracture strength compared with those of the Ho sample. Usually, increasing the yield strength and fracture strength of a material requires sacrificing its ductility. However, we achieved an increase in fracture strength without compromising the plasticity of the material. The TEM technique was used to better characterize the microstructures of the Ho $\text{Fe}_{50}\text{Mn}_{27}\text{Ni}_{10}\text{Cr}_{13}$ alloy and HPHT-treated Fe-based alloy samples. The corresponding selected area electron diffraction (SAED) patterns of the grains suggest that only the FCC phase exists in the Ho $\text{Fe}_{50}\text{Mn}_{27}\text{Ni}_{10}\text{Cr}_{13}$ sample (Fig. R7b). The high-resolution TEM image reveals a grain boundary structure with a transition layer between neighboring grains in the Ho $\text{Fe}_{50}\text{Mn}_{27}\text{Ni}_{10}\text{Cr}_{13}$ sample (Fig. R7c). Compared to the Ho $\text{Fe}_{50}\text{Mn}_{27}\text{Ni}_{10}\text{Cr}_{13}$ sample, the grain boundary seems to become smoother in the HPHT-treated $\text{Fe}_{50}\text{Mn}_{27}\text{Ni}_{10}\text{Cr}_{13}$ sample, as shown in Fig. R7d. The high-resolution TEM image (Fig. R7e) reveals an extremely sharp grain boundary in the HPHT-treated $\text{Fe}_{50}\text{Mn}_{27}\text{Ni}_{10}\text{Cr}_{13}$ sample. The corresponding selected area electron diffraction (SAED) pattern shows that the two grains have a specific orientation relationship of $(11\bar{1})\parallel(200)$, which meets the classical Kurdjumov-Sachs (KS) relationship (T. Xiong et al., *J. Mater. Sci. Technol.* 2021;65;216-227). The formation of such grain boundary is energetically favorable and may be beneficial to improve the mechanical properties. These results thus provide strong evidence that the HPHT strategy proposed in the present work is not only limited to eutectic high entropy alloys or pure copper, but is promising to be implemented for other alloy systems.

Figure R7 a, Engineering stress-strain curves of the HPHT-treated $\text{Fe}_{50}\text{Mn}_{27}\text{Ni}_{10}\text{Cr}_{13}$ sample compared to that of the Ho $\text{Fe}_{50}\text{Mn}_{27}\text{Ni}_{10}\text{Cr}_{13}$ sample, showing a synergistic enhancement of both strength and ductility. **b**, Low magnification TEM image of the Ho $\text{Fe}_{50}\text{Mn}_{27}\text{Ni}_{10}\text{Cr}_{13}$ sample, showing a transition layer at the interface. The insets show related SAED patterns of the grains. **c** Typical HRTEM image of the Ho $\text{Fe}_{50}\text{Mn}_{27}\text{Ni}_{10}\text{Cr}_{13}$ sample, which demonstrates the incoherent interface. **d**, Low magnification TEM image of the HPHT-treated $\text{Fe}_{50}\text{Mn}_{27}\text{Ni}_{10}\text{Cr}_{13}$ sample. The

insets show related SAED patterns of the interface region, which meets the K-S relationship. e, Typical HRTEM image and the corresponding IFFT image of the HPHT-treated $\text{Fe}_{50}\text{Mn}_{27}\text{Ni}_{10}\text{Cr}_{13}$ sample, which demonstrates a coherent interface.

Regarding the effect of HPHT on the microstructure and mechanical properties of pure copper, it can be seen that strength strongly increased significantly without the expense to the ductility in the HPHT-treated Cu sample. Although the ductility does not increase as much as strength, the HPHT-treated Cu alloy actually exhibits an exceptional combination of high strength and large uniform elongation, in comparison with the enhanced mechanical properties observed in the well-known gradient nano-grained Cu sample (K. Lu et al., Science 2011;331;1587), as shown in Fig. R8.

Overall, the newly performed experiments, combined with our previous results, have demonstrated that the HPHT treatment can be a promising paradigm for achieving strength-ductility synergy in metals and alloy systems. We have carefully revised the manuscript and the supplementary materials to clarify this point.

Figure R8 Engineering stress-strain curves of the Cu-HPHT sample compared to that of the Cu-Ho sample and the gradient nano-grained Cu sample.

3) For the Al-containing CoCrFeNi HEAs, numerous studies have been performed with respect to microstructure control to enhance strength-ductility combination via grain refinement or hetero-structuring for past decade.

Reply: We appreciate the reviewer for this comment. We agree with the reviewer that the control of microstructure to enhance strength-ductility combination for $\text{Al}_x\text{CoCrFeNi}$ high entropy alloys have been attracting numerous attentions in recent years. Although various kinds of processing techniques and microstructural control studies have been developed to enhance their plastic deformation capacities, the essential problem behind its brittleness remains unresolved due to the quite limited freedom for microstructural adjustments during traditional manufacturing processes, even with the design of heterogeneous microstructures (T. Xiong et al., Scr. Mater.

2020;186;336-340. P.J. Shi et al., Nat. Comm. 2019;10;489. T.H. Wang et al., J. Alloys Compd. 2018;766;312-317. X.Z. Gao et al., Acta Mater. 2017;141;59-66. I.S. Wani et al., Mater. Res. Lett. 2016;4;174-179). To overcome such problem, recent studies have been carried out to investigate the capability of advanced and versatile processing techniques, such as additive manufacturing (J. Ren et al., Nature 2022;608;62-68). Although many remarkable achievements have been obtained, there is still a relentless quest to achieve significantly enhanced mechanical properties.

The processing technique with the combination of high pressure and high temperature proposed in the present work offers a great prospect for altering the microstructure of structural materials that would permit overcoming the strength-ductility trade-off. Given the vast space of control parameters involved in the HPHT treatment and the often complex microstructural evolution under extreme pressures and temperatures, the novel mechanical behaviors of HPHT-treated structural materials are intriguing to investigate, especially the abnormal kinetics and thermodynamics of non-equilibrium interfaces induced by the HPHT treatment. A comprehensive understanding of the process-microstructure-mechanical properties relationship with HPHT is also of scientific importance. More importantly, the processing conditions of high pressure and high temperature proposed in the present work for large-volume structural materials have never been reported in the literature. By using a hexahedron-anvil-press apparatus, we have demonstrated that eutectic high entropy alloys can be transformed to having a unique hierarchically patterned microstructure with coherent interfaces and simultaneously doubling its strength to 1.15 gigapascals and enhancing its tensile ductility to 36% after the HPHT treatment, overcoming the typical brittle interfacial fracture mode in conventional eutectic high entropy alloys. Such unusual HPHT-induced strength-ductility synergy is attributed to the formation of a hierarchically patterned FCC/BCC microstructure with coherent interfaces, which promotes the activation of multiple deformation mechanisms at multiple length scales.

Overall, the proposed HPHT strategy can engineer microstructures to promote uniform plastic deformation and prevent strain localization, which has so far not been applied to Al-containing CoCrFeNi HEAs or other alloy systems. In this perspective, our results point out a promising pathway for developing high-performance structural materials.

4) In order to emphasize that the HPHT process is superior for enhancing mechanical of the eutectic HEAs, the authors should investigate the process-induced microstructure change in not only Al_{0.7}CoCrFeNi HEA but also AlCoCrFeNi_{2.1}, the most famous eutectic HEA [Nat. Comm. 10 (2019) 489, Nature 608 (2022) 62-68], or other eutectic HEAs. This will strongly support the versatility and usefulness of this process for improving properties for the eutectic HEA.

Reply: We sincerely thank the reviewer for this helpful suggestion. As a newly emerged field in

materials science, eutectic high entropy alloys (EHEAs) offer a rich playground for designing novel materials for structural applications because of their huge compositional ranges and promising mechanical properties. The optimal design of EHEAs is indeed complicated since it must be considered comprehensively about many critical issues including both the microstructures and properties.

First, we would like to explain the reason of choosing $\text{Al}_{0.7}\text{CoCrFeNi}$ HEA in the original manuscript. For the $\text{AlCoCrFeNi}_{2.1}$ system, the as-cast sample itself has a good combination of strength and ductility. Many previous studies have shown that a remarkable strength-ductility enhancement can be achieved via an appropriate thermal and mechanical treatment. Besides, the $\text{Al}_x\text{CoCrFeNi}$ system is prominent among several alloy systems identified on this alloy-design strategy. The $\text{Al}_x\text{CoCrFeNi}$ system has been used to elucidate their mechanical and anticorrosive properties. However, existing experimental studies on brittle $\text{Al}_{0.7}\text{CoCrFeNi}$ are quite limited. Compared with $\text{AlCoCrFeNi}_{2.1}$ alloy, the brittle interfacial fracture is more obvious in $\text{Al}_{0.7}\text{CoCrFeNi}$ alloy. Consequently, it is meaningful to explore the microstructural evolution and mechanical properties of the $\text{Al}_{0.7}\text{CoCrFeNi}$ alloy. Therefore, we chose the brittle $\text{Al}_{0.7}\text{CoCrFeNi}$ alloy as the model to demonstrate our HPHT strategy. Based on our results, we expect that the HPHT treatment can play a great role in regulating the microstructure and improving the mechanical properties of the structural materials in the future.

Second, following the reviewer's valuable suggestion, we have now performed additional experiments using the HPHT treatment engineering protocol for the $\text{AlCoCrFeNi}_{2.1}$ alloy. As shown in Fig. R9, the HPHT-treated $\text{AlCoCrFeNi}_{2.1}$ alloy exhibits improved fracture strength and ductility compared with those of the Ho $\text{AlCoCrFeNi}_{2.1}$ alloy. In addition, the dual-phase lamellae structure frequently observed in the Ho alloy can be transformed into a hierarchically patterned microstructure. Although the effect of HPHT treatment is related to alloy composition, initial atomic structure and processing conditions, we have demonstrated that the HPHT treatment can help overcome the strength-ductility trade-off in various eutectic HEAs. Our findings are calling for future work on comprehensive studies of the discovery of novel high-performance HPHT-treated alloys.

We have now added these new results in the revised manuscript support the versatility and usefulness of HPHT process for improving properties for the eutectic HEA.

Figure R9 a, Engineering stress-strain curves of the HPHT-treated $\text{AlCoCrFeNi}_{2.1}$ sample compared to that of the Ho $\text{AlCoCrFeNi}_{2.1}$ sample. **b**, The corresponding EBSD phase map reveals that the dual-phase lamellae consist of alternating FCC layers and BCC layers in the Ho alloy. **c**, The corresponding EBSD phase map reveals that the dual-phase lamellae structure has been modified in the HPHT-treated alloy.

5) It has been well known that the mechanical properties of the as-cast or as-homogenized ingot are lower than that of the samples treated by thermo-mechanical process (i.e., forging, heat treatment) due to coarse grain structure of the ingots. The authors should show the tensile curves of not only as-homogenized ingot but also the sample treated by conventional thermo-mechanical process for emphasizing the properties of the sample treated by HPHT.

Reply: Following the reviewer's suggestion, we have tested the tensile properties of the $\text{Al}_{0.7}\text{CoCrFeNi}$ alloy treated by conventional processes. Specially, the samples for microstructural and tensile studies were obtained from the cold-rolled sheets and subsequently annealed at 1273 K for 1 hour to obtain a recrystallized microstructure. The tensile stress-strain curves of the specimens in the homogenized (Ho), cold-rolled (CR) and annealed conditions (CR+Annealed) are shown in Fig. R10. Cold-rolling results in a drastic increase in strength at the expense of elongation. The CR+Annealed sample shows improved strength with reduced ductility. These methods are similar to conventional methods for strengthening materials, which invariably suffer from the adverse consequence that the increase in strength facilitated by interactions between internal defects also causes reduced ductility.

In the revised manuscript, we have now added these new results in the Supplementary Information.

Figure R10 True stress-strain curves of the Ho alloy, along with the cold-rolled alloy and the CR+Annealed alloy.

Minor comments.

1. The global chemical composition of the alloy should be presented. Moreover, the chemical composition of each phase (i.e., FCC and B2) should be presented.

Reply: We sincerely thank the reviewer for the suggestion. Following the reviewer's comment, we have now presented the global chemical composition of the alloy and the chemical composition of the two phases in the revised manuscript.

2. The regions at different local strains for EBSD analysis of the deformed sample should be indicated for the readers.

Reply: We have indicated the regions at different local strains for EBSD analysis of the deformed sample following your comment in Supplementary Fig. 12 and Supplementary Fig. 16.

3. In order to investigate GND evolution in the sample during deformation, the EBSD analysis in higher magnification is needed for higher accuracy. Moreover, the authors should present EBSD GND maps, not KAM maps.

Reply: We thank the reviewer for this constructive comment. We fully agree with you that, as an averaged estimation from KAM, the area used for the calculation of GND should not be too small. Therefore, the current EBSD magnification has been carefully ensured to show the unique hierarchically patterned microstructure. It can be observed that the featured hierarchically patterned microstructure is well preserved during loading, which can even be identified on the fracture surface after the extended uniform elongation, indicating the high structural stability of the

HPHT-treated alloy. In addition, the accuracy of the analysis is related to the step size for the EBSD test. Although higher accuracy can be obtained with a smaller step size, the test becomes more expensive and time-consuming. We have chosen a step size of 0.1 μm for the analysis, which is sufficient to observe the unique microstructure in our HPHT sample.

As shown in Fig. R11 and Fig. R12, we have provided the GND maps in the Ho and HPHT-treated alloys at different strain amplitudes. In the Ho alloy, the B2 phase possesses much smaller GND values than the FCC phase in the later stage of tensile deformation. A relatively homogeneous distribution of GND is observed at the final fracture with a plastic strain of 35%, contributing to the extra tensile elongation of the HPHT alloy. These observations are consistent with our KAM analysis. The GND value is estimated and calculated from the KAM value. The KAM value is measured to reveal the deformation-induced local orientation gradients inside the grains. The area with relatively higher KAM indicates severe stress localization features. Note that many previous studies have used the EBSD-KAM maps to show the distributions of misorientations (D. Raabe et al. Nature 2022;608;301-316. Y.M. Wang et al., Nat. Mater. 2018;17;63-71. S.B. Gao et al., Nat. Commun. 2023;14;6752. Y.L. Zhao et al., Acta Mater. 2022;223;117480. S.S. Sohn et al., Adv. Mater. 2019;31;1807142).

Following the reviewer's suggestion, we have now added GND map results in the revised Supplementary Information.

Figure R11 a, The corresponding phase distribution maps of the Ho alloy at the strain amplitudes of 3%, 7%, and 15%. **b**, The corresponding geometrically necessary dislocations (GND) maps.

Figure R12 a, The corresponding phase distribution maps of the HPHT-treated alloy at the strain amplitudes of 3%, 7%, and 15%. **b**, The corresponding geometrically necessary dislocations (GND) maps.

4. As shown in Fig. 2(a), the strain hardening ability of the HPHTed sample is similar with that of as-homogenized sample. Moreover, it seems that the strain hardening rate of Ac+HPHTed sample is higher than Ho-HPHTed sample. The strain hardening rate is highly related to strength-ductility combination of metallic materials. The authors should discuss the strain hardening behaviors of the samples.

Reply: We fully agree with the reviewer that the work hardening behaviors of the alloys should be discussed. As shown in Fig. R13a, the tensile true stress-strain curves of the Ac, Ac+HPHT, Ho and Ho+HPHT alloys exhibit remarkable differences in fracture stress and strain. The Ac+HPHT alloy exhibits a remarkable improvement in fracture true strain compared to the Ac alloy, while the fracture true strength is almost the same. The Ho alloy only has a fracture true strain of $\sim 14.7\%$. The HPHT alloy doubles the fracture strain to $\sim 30.7\%$. More importantly, the HPHT alloy exhibits a high fracture true stress of ~ 1570 MPa, which is much higher than those of the AC, Ac+HPHT and Ho alloys, respectively.

The corresponding work hardening rate curves of these alloys have been calculated, as shown in Fig. R13b. The Ac alloy exhibits a continuous decrease tendency. The Ac+HPHT alloy shows a certain work hardening capability. The strain hardening of Ho alloy does not decrease as fast as that of Ac and Ac+HPHT alloys. The work-hardening capability of HPHT-treated alloy is much more pronounced and stable until reaching the tensile plastic instability region. The different strain-hardening rate curves demonstrate the occurrence of different deformation behaviors during plastic deformation. These results suggest that the deformation mechanisms triggered in the HPHT-treated alloy have a higher capability to induce strain hardening, hence leading to the

excellent combination of strength and ductility.

In the revised manuscript, following the reviewer's suggestion, we have added the above results and discussion.

Figure R13 a, True stress-strain curves of the homogenized alloy (Ho alloy) after HPHT treatment, along with those of the Ho alloy before HPHT treatment and the as-cast alloy (Ac alloy) with and without HPHT treatment. **b**, Strain-hardening rate with respect to true strain corresponding to the engineering stress-strain curves shown in a.

Reviewer #3

Developing unique structures with substantial improvement of both strength and ductility is highly challenging. Heterostructures materials have been recently reintroduced, among them so-called harmonic structures. However, such a methodology has yet to be utilized on high entropy alloys, especially on eutectic HEAs. Indeed, the absence of characterization of these alloys' more or less fragile behavior has already slowed down attempts to develop HEA alloys as structural materials. From this point of view, the study presented here is an approach that could benefit the community. The analysis is well-detailed, and the process engineering approach is attractive, clear, and easily reproduced by the interested community. Below are some comments that the authors could comment on/discuss:

Reply: We sincerely thank the reviewer for giving positive comments on our work. We have carefully addressed your comments in the following responses.

1) This study lacks statistics and quantification, which raises the question of whether the observed effects and the underlying mechanisms are anecdotal or local (regarding observations at very fine scales). The absence of this analysis would remove all the interests of the discussion.

Reply: We sincerely thank the reviewer for this valuable comment. To address the reviewer's concern on the underlying deformation mechanisms of the HPHT-treated alloy and Ho alloy, we have carried out five individual ECCI analyses for different locations at the same strain amplitude to confirm the difference in the observed deformation characteristics of the HPHT-treated alloy (Fig. R14) and Ho alloy (Fig. R15). At a low strain, dislocations can be observed in the FCC and B2 phases at different locations for the HPHT-treated alloy (Fig. R14a-c). With increasing strain, dislocation density increases in both phases, leading to the formation of microbands and high-density dislocation walls near the interface (Fig. R14d-f). At a later stage of deformation, deformation twins are activated to further accommodate deformation, subdividing microbands and promoting mutual interactions between refined microbands (Fig. R14g-i). In comparison, for the Ho alloy, few dislocations were observed in the FCC phase and no dislocations in the B2 phase at a low strain (Fig. R15a-c). With increasing strain, stacking faults were activated in the FCC phase and almost no dislocation in the B2 phases (Fig. R15d-f). At a later stage of deformation, the limited slip lines only exist in the FCC phase while the BCC phase exhibits no evident dislocation activities. Dislocation pile-ups were evident around the fine B2 phase (Fig. R15g-i). Note that there was no evidence of high-density dislocation walls and deformation twins at different locations in the Ho alloy, while high-density dislocation walls and deformation twins can be observed in the HPHT-treated alloy. Our TEM analysis of the fracture region further demonstrates slip transfer between the FCC and B2 phases, numerous dislocation-stacking faults interactions and deformation

twins in the HPHT-treated alloy (Fig. 4 in the manuscript), in distinct contrast to the localized deformation at the interface and the absence of dislocation in the B2 phase in the Ho alloy (Fig. 10 in the Supplementary Information). These results have demonstrated the activation of multiple deformation mechanisms in the HPHT-treated alloy, including dislocations, stacking faults, microbands and deformation twins. Such progressive activation of multiple plastic deformation mechanisms in the HPHT alloy is in sharp contrast to the case of the alloy without HPHT treatment.

The quantification of deformation structures at very fine scales by ECCI has been a challenge, especially for dual-phase alloy systems. (Z.M. Li et al., *Nature* 2016;534;17981. T. Yang et al., *Science* 2018;362; 933-937. P.J. Shi et al., *Nat. Commun.* 2019;10;489. L.L. Han et al., *Adv. Mater.* 2021;33;2102139). Nevertheless, this does not detract from the importance of our findings, as we have confirmed the distinct deformation mechanisms of the HPHT-treated alloy and Ho alloy by performing a series of individual ECCI analyses. More importantly, these deformation characteristics at fine scales of the HPHT-treated and Ho alloys are consistent with their respective crack behaviors. The elongated hollows with large widths along interfaces appeared in the Ho alloy, which indicates that interface-induced cracks dominate the eventual fracture process. Isolated microcracks are uniformly distributed near the fracture surface in the HPHT-treated alloys, the microcracks can cut through the FCC and B2 phases in the HPHT-treated alloy, indicating that high local stresses were shielded and plastic deformation was better accommodated. Also, the B2 phase with high stability can isolate the microcracks and prevent microcracks from coalescing into elongated voids in the HPHT-treated alloy. Furthermore, these two phases can deflect crack paths, thereby delaying the crack extension and hence suppressing the cracking in the HPHT-treated alloy. In addition, our macro EBSD-KAM analysis has also demonstrated the distinct deformation behaviors in the HPHT-treated alloy and the Ho alloy. The increased misorientation distribution of FCC and B2 phases with increasing strain indicates strain delocalization in the HPHT-treated alloy (Fig. 3 in the manuscript). In the Ho alloy, however, the misorientation of the B2 phase shows a severe mismatch with that of the FCC phase, indicating a highly incompatible plastic strain distribution.

Overall, the fine-scale ECCI analyses, combined with the macroscale fracture behavior observations and EBSD-KAM analyses, have revealed a novel synergic deformation mechanism in HPHT-treated alloys: the activation of multiple deformation mechanisms, including SFs, microbands and deformation twins, endowing the HPHT-treated alloy with excellent structural stability and strain-hardening capability. To address the reviewer's concern, we have added additional results, discussion and explanations in the revised Supplementary Information to clarify this point.

Figure R14 Electron contrast channel imaging (ECCI) analyses reveal the evolution of the deformation substructure in the HPHT-treated alloy at strain amplitudes of 5%, 15%, and 35%. Five different locations of the fractured sample (scale bar is 0.35 mm) were measured at the same strain amplitude.

Figure R15 Electron contrast channel imaging (ECCI) analyses reveal the evolution of the

deformation substructure in the Ho alloy at strain amplitudes of 3%, 7%, and 15%. Five different locations of the fractured sample (scale bar is 0.35 mm) were measured at the same strain amplitude.

2) A rather critical point concerns using KAM to estimate the density of GNDs. Regarding the EBSD data, it is trendy to use an equation immediately if it has been used somewhere in a paper. But while KAM is essentially a scalar number, the sought GND density is tensor and should be determined as such. So, computing dislocations density via KAM is generally wrong, and this becomes obvious in the case of an elastically bent crystal domain with no dislocations. Indeed, the proposed treatment of EBSD data will give a senseless non-zero result here. A few interesting investigations from EDAX-TSL remarkably showed that a different step size results in different GNDs. How trustful can GNDs be if it depends on the acquisition conditions? This has a significant impact on the whole discussion that follows. Notice that techniques such as Convolutional Multiple Whole Profile fitting of DRX patterns are more efficient and robust.

Reply: We sincerely thank the reviewer for raising these comments and fully agree with the reviewer's concern on the estimation of GND density. Indeed, it is challenging to measure well-targeted quantitative dislocation density. Before showing our additional experimental results using the method suggested by the reviewer, we would like to point out that the KAM value has been widely used in the literature due to its feasibility of revealing deformation-induced local orientation gradients inside the grains and the storage of geometrically necessary dislocations (GNDs). In other words, the KAM value is associated with the density of geometrically necessary dislocations (GNDs) in the detected region, i.e., higher KAM values suggest a higher density of GNDs and more plastic strains in these zones. Specifically, the method to evaluate GND densities follows Kubin and Mortensen (L.P. Kubin et al., Scr. Mater. 2003; 48; 119-125), they defined a GND array for simple cylinder torsion based on the strain gradient model by Gao et al. (H. Gao et al., J. Mech. Phys. Solids 1999;47;1239-1263). Assuming a series of twist subgrain boundaries in the cylinder, each containing two perpendicular arrays of screw dislocations, the misorientation angle θ is related to the GND density ρ , $\theta = \rho b$, where u is the unit length vector and b is the Burger's vector. As a first order approach, KAM is retrieved directly from EBSD data, which is chosen as a measure for the local misorientations. The KAM quantifies the average misorientation around a measurement point with respect to a defined set of nearest or nearest plus second-nearest neighbor points. It has been demonstrated that this method is appropriate for calculating GND densities from EBSD data sets (D. Raabe et al., Materials Science and Engineering A 2010;527;2738-2746). Therefore, many studies have used the measured KAM (D. Raabe et al., Nature 2022;608;301-316. Y.M. Wang et al., Nat. Mater. 2018;17;63-71) and the

estimated GND (S.B. Gao et al., Nat. Commun. 2023;14;6752. Z.M. Li et al., Acta Mater. 2017;131;323-335) to show the distributions of deformation-induced misorientations.

We fully agree with the reviewer that the step size can influence the statistics obtained from the KAM-based calculation method, meaning that the GND calculated from the KAM should be viewed as an estimation of the dislocation density, which helps us understand the difference in deformation behaviors of the HPHT-treated alloy and Ho alloy. In the present work, we have used the same step size of 0.1 μm for the HPHT-treated and Ho alloy. The step size is smaller than that commonly used in the reported work (S.B. Gao et al., Nat. Commun. 2023;14;6752. Q.S. Pan et al., Acta Mater. 2023;244;118542. Y.M. Wang et al., Nat. Mater. 2018;17;63-71. D.D. Zhang et al., Acta Mater. 2022;233;117981). The smaller the step size, the higher the resolution, but at the same time, the actual time cost will be greatly increased. With this step size, we have shown that the estimated GND densities in the HPHT alloy are much higher than those in the Ho alloy, indicating improved capability of dislocation storage in the HPHT-treated alloy. Moreover, the EBSD-KAM-GND analysis has demonstrated the homogeneous deformation in the HPHT-treated alloy in contrast to the localized deformation in the Ho alloy.

Following the reviewer's valuable suggestion, we have now performed additional experiments to take into account the total dislocation density. The dislocation density can be roughly estimated from the XRD patterns by employing the Williamson-Hall method (G.K. Williamson, W.H. Hall, Acta Metall. Mater. 1953;1;22-31. D. Raabe et al. Nature 2022;608;301-316). The XRD patterns of the tensile-deformed Ho and HPHT-treated alloy at various strain amplitudes are presented in Fig. R16.

Figure R16 The X-ray diffraction patterns of the Ho and HPHT-treated alloys at different strain amplitudes.

Figure R17 a, Dislocation density against strain in the FCC and the B2 phases for the Ho alloy. **b**, Dislocation density against strain in the FCC and the B2 phases for the HPHT-treated alloy.

The calculated dislocation densities are illustrated in Fig. R17. As shown in Fig. R17a, of the Ho alloy increases from $2.68 \times 10^{14} \text{ m}^{-2}$ to $3.75 \times 10^{14} \text{ m}^{-2}$ from 0% to 3% strains, whereas the increases from $7.51 \times 10^{13} \text{ m}^{-2}$ to $1.81 \times 10^{14} \text{ m}^{-2}$. Upon further loading, the dislocation density of the FCC phase increases to $9.79 \times 10^{14} \text{ m}^{-2}$ at 15% strain, which is higher than that of the B2 phase. In the HPHT-treated alloy (Fig. R17b), the dislocation density of the FCC phase increases rapidly from $2.86 \times 10^{14} \text{ m}^{-2}$ to $6.99 \times 10^{14} \text{ m}^{-2}$ from 0% to 15% strains, whereas the dislocation density of the B2 phase increases from $1.03 \times 10^{14} \text{ m}^{-2}$ to $1.04 \times 10^{15} \text{ m}^{-2}$. Upon further loading, the dislocation density of the B2 phase increases to $1.91 \times 10^{15} \text{ m}^{-2}$ at 35% strain, whereas the dislocation density of the FCC phase increases to $1.63 \times 10^{15} \text{ m}^{-2}$ at 35% strain. These results confirm the uniform deformation of the HPHT-treated alloy. In addition, the dislocation densities in the HPHT alloy are much higher than those in the Ho alloy, again suggesting the improved capability of dislocation storage in the HPHT-treated alloy.

We have now added the above explanations and additional experimental results in the revised manuscript to support our discussion.

3) It is said that different properties can be generated by varying the experimental conditions. The supplementary Fig. 5, which is supposed to reflect this assertion, must be more conclusive. Without any quantification, the two figures are very close regarding mechanical characteristics and appearance. Did I miss something?

Reply: We sincerely thank the reviewer for raising this comment. With Supplementary Fig. 5 (Fig. R18), we intended to say that the microstructure of the HPHT-treated alloys, i.e., the variation and distribution of two phases, can be tuned by the pressure and temperature conditions.

Figure R18 a, Electron backscatter diffraction (EBSD) inverse-pole figure (IPF) maps in the $\text{Al}_{0.7}\text{COCrFeNi}$ alloy treated at 6 GPa and 1373 K. **b**, Corresponding EBSD phase maps in the $\text{Al}_{0.7}\text{COCrFeNi}$ alloy treated at 6 GPa and 1373 K. **c**, Engineering stress-strain curve of the $\text{Al}_{0.7}\text{COCrFeNi}$ alloy treated at 6 GPa and 1373 K. **d**, Electron backscatter diffraction (EBSD) inverse-pole figure (IPF) maps in the $\text{Al}_{0.7}\text{COCrFeNi}$ alloy treated at 6 GPa and 1473 K. **e**, Corresponding EBSD phase maps in the $\text{Al}_{0.7}\text{COCrFeNi}$ alloy treated at 6 GPa and 1473 K. **f**, Engineering stress-strain curve of the $\text{Al}_{0.7}\text{COCrFeNi}$ alloy treated at 6 GPa and 1473 K. **g**, Electron backscatter diffraction (EBSD) inverse-pole figure (IPF) maps in the $\text{Al}_{0.7}\text{COCrFeNi}$ alloy treated at 6 GPa and 1573 K. **h**, Corresponding EBSD phase maps in the $\text{Al}_{0.7}\text{COCrFeNi}$ alloy treated at 6 GPa and 1573 K. **i**, Engineering stress-strain curve of the $\text{Al}_{0.7}\text{COCrFeNi}$ alloy treated at 6 GPa and 1573 K.

In addition to the HPHT-treated alloy treated at 1473 K, we also treated the alloy at 1373 K and 1573 K, which are denoted as HPHT-treated-1373K and HPHT-treated-1573K, respectively. The corresponding electron backscatter diffraction (EBSD) phase map of the HPHT-treated-1373 K alloy reveals a phase arrangement pattern in which the polygon structural units are interconnected by the FCC phase. The volume fractions of FCC and BCC phases in the HPHT-treated-1373 K alloy were determined by EBSD to be ~ 69% and ~ 31%, identical to those in the Ho alloy and HPHT-treated-1473K alloy (~69% and ~31%). With the optimal temperature of HPHT treatment, the corresponding microstructure transforms into a unique hierarchically patterned microstructure. Although the mechanical properties of the HPHT-treated-1373K alloy are not as good as the HPHT-treated-1473K alloy, the specific structure caused by HPHT can already provide substantial property enhancement compared to the Ho alloy. For the HPHT-treated-1573K alloy, obvious grain growth and phase segregation can be seen in Fig. R18. The volume fractions of FCC and BCC phases in the HPHT-treated-1573K alloy were determined by EBSD to be ~ 58% and ~ 42%. The segregation and the high content of the BCC phase will lead to a decrease in mechanical properties. Although the mechanical properties have slightly decreased compared with those of HPHT-treated-1473K alloy, they are significantly improved in comparison with those of the Ho alloy. These results indicate that the HPHT conditions can be tuned to modify the microstructure of the Al_{0.7}CoCrFeNi alloy.

In the revised manuscript, we have added sentences to explain these points more clearly.

REVIEWERS' COMMENTS

Reviewer #1 (Remarks to the Author):

Thank you for the detailed responses. All of my comments have been addressed and I am satisfied with the revised version. I recommend the publication of this manuscript in Nature Communications. There are a few minor comments:

- 1) The unit of force should only be specified in GPa or MPa (MPa would be preferred) in the full paper.
- 2) I would present the Fig. R4 (or Supplementary Fig. 7a) instead of Fig. 2a in the paper, showing the true stress which can be reached of 1570 MPa in the HPHT sample. In this case, the corresponding ("This work") data in Fig. 2b could be shifted to higher level of 1570 MPa. In the revised version, the Author have already mentioned the high fracture true stress of 1570 MPa (see line 144 of pdf file), but it cannot be read in Figs. 2a and 2b.
- 3) The AC designation should be Ac (see line 145).
- 4) There is an abbreviation in line 315, as "EHEA", but its meaning has not been given. I think it is "Eutectic High Entropy Alloys". Please fix this problem.
- 5) Line 340: it should say: (step size = 0.1 μm) (not 0.1 um).

Reviewer #2 (Remarks to the Author):

The authors successfully addressed the reviewer's comments, and the reviewer suggests that the paper can be accepted after correction of minor typo errors (e.g., "~~ homogenized Al_{0.7}COCrFeNi ~~").

Reviewer #3 (Remarks to the Author):

The authors have conscientiously responded to the reviewer's questions by providing complete and relevant clarifications, which allows me to recommend this work to the editor for publication.

Point-by-point Response to Reviewer Comments

RE: NCOMMS-23-46785A

Title: Overcoming strength-ductility tradeoff with high pressure thermal treatment

We highly appreciate the reviewers' constructive comments and valuable suggestions on our manuscript. Based on these comments, we have carefully revised the manuscript. In the following, the review comments are listed in *italic* blue font and our response to each comment is given in **black** font.

Reviewer #1

Thank you for the detailed responses. All of my comments have been addressed and I am satisfied with the revised version. I recommend the publication of this manuscript in Nature Communications.

Reply: We deeply thank the reviewer for the positive comments on our revisions.

There are a few minor comments:

1) The unit of force should only be specified in GPa or MPa (MPa would be preferred) in the full paper.

Reply: We have specified the unit of force in MPa in the full paper.

2) I would present the Fig. R4 (or Supplementary Fig. 7a) instead of Fig. 2a in the paper, showing the true stress which can be reached of 1570 MPa in the HPHT sample. In this case, the corresponding ("This work ") data in Fig. 2b could be shifted to higher level of 1570 MPa. In the revised version, the Author have already mentioned the high fracture true stress of 1570 MPa (see line 144 of pdf file), but it cannot be read in Figs. 2a and 2b.

Reply: We deeply appreciate the reviewer's good suggestion. We have presented the true stress-strain curves in the inset of Fig. 2a, showing the true stress which can be reached of 1570 MPa in the HPHT sample.

3) The AC designation should be Ac (see line 145).

Reply: We appreciate the careful reading of the manuscript. We have changed 'AC' to 'Ac'.

4) *There is an abbreviation in line 315, as “EHEA”, but its meaning has not been given. I think it is “Eutectic High Entropy Alloys”. Please fix this problem.*

Reply: We have defined the ‘EHEA’ as the reviewer recommended.

5) *Line 340: it should say: (step size = 0.1 μm) (not 0.1 um).*

Reply: We have corrected the mistake in the revised manuscript.

Reviewer #2

The authors successfully addressed the reviewer's comments, and the reviewer suggests that the paper can be accepted after correction of minor typo errors (e.g., “~~ homogenized Al_{0.7}COCrFeNi ~~”).

Reply: We deeply thank the reviewer for the positive comments on our revisions. We have revised ‘homogenized Al_{0.7}COCrFeNi ’ to ‘homogenized Al_{0.7}COCrFeNi ’.

Reviewer #3

The authors have conscientiously responded to the reviewer's questions by providing complete and relevant clarifications, which allows me to recommend this work to the editor for publication.

Reply: We sincerely thank the reviewer for his/her positive comments.